# RNA inhibitors of nuclear proteins responsible for multiple organ dysfunction syndrome

Kevin T. Urak[1,2], Giselle N. Blanco[1], Shambhavi Shubham[1], Li-Hsien Lin[1], Justin P. Dassie[1], William H. Thiel[1,3], Yani Chen[1], Vijay Kumar Sonkar[1], Beilei Lei[4], Shubha Murthy[1], Wade R. Gutierrez [5], Mary E. Wilson[1,6,7], Jonathan A. Stiber[4], Julia Klesney-Tait[1], Sanjana Dayal[1], Francis J. Miller Jr [4,8,9] & Paloma H. Giangrande[1,2,3,10,11,12,13]

The development of multiple organ dysfunction syndrome (MODS) following infection or tissue injury is associated with increased patient morbidity and mortality. Extensive cellular injury results in the release of nuclear proteins, of which histones are the most abundant, into the circulation. Circulating histones are implicated as essential mediators of MODS. Available anti-histone therapies have failed in clinical trials due to off-target effects such as bleeding and toxicity. Here, we describe a therapeutic strategy for MODS based on the neutralization of histones by chemically stabilized nucleic acid bio-drugs (aptamers). Systematic evolution of ligands by exponential enrichment technology identified aptamers that selectively bind those histones responsible for MODS and do not bind to serum proteins. We demonstrate the efficacy of histone-specific aptamers in human cells and in a murine model of MODS. These aptamers could have a significant therapeutic benefit in the treatment of multiple diverse clinical conditions associated with MODS.

[1] Internal Medicine, University of Iowa, Iowa City, IA 52242, USA. [2] Molecular & Cellular Biology Program, University of Iowa, Iowa City, IA 52242, USA. [3] Abboud Cardiovascular Research Center, University of Iowa, Iowa City, IA 52242, USA. [4] Department of Medicine, Duke University, Durham, NC 27708, USA. [5] Medical Scientist Training Program, University of Iowa, Iowa City, IA 52242, USA. [6] Department of Microbiology, University of Iowa, Iowa City, IA 52242, USA. [7] Veteran's Affairs Medical Center, University of Iowa, Iowa City, IA 52241, USA. [8] Pharmacology and Cancer Biology Program, Duke University, Durham, NC 27708, USA. [9] Deptartment of Medicine, Veterans Administration Medical Center, Durham, NC 27705, USA. [10] Interdisciplinary Graduate Program in Genetics, University of Iowa, Iowa City, IA 52242, USA. [11] Holden Comprehensive Cancer Center, University of Iowa, Iowa city, IA 52242, USA. [12] Radiation Oncology, University of Iowa, Iowa City, IA 52242, USA. [13] Environmental Health Sciences Research Center (EHSRC), University of Iowa, Iowa City, IA 52242, USA. These authors contributed equally: Kevin T. Urak, Giselle N. Blanco, Shambhavi Shubham. Correspondence and requests for materials should be addressed to F.J.M.Jr. (email: francis.miller@duke.edu) or to P.H.G. (email: paloma-giangrande@uiowa.edu)

Approximately 45% of patients who develop multiple organ dysfunction syndrome (MODS) will die due to acute secondary organ injury/failure[1]. Survivors are at risk of developing persistent mental and physical impairments. MODS occurs after a severe cytotoxic insult such as sepsis, trauma, ischemia/reperfusion injury, pancreatitis, peritonitis, stroke, thrombosis and autoimmune disease[2–4]. MODS is characterized by the release of molecular mediators from damaged tissues which create a domino effect including capillary leak, interstitial edema, hemorrhage and systemic inflammation[5]. MODS is primarily managed with supportive care, as there is no approved treatment to prevent or reverse it. The realization that damaged cells release their nuclear content into the circulation suggests nuclear proteins as potential therapeutic targets for MODS[2,6]. Since histones are the most abundant proteins in the nucleus, they have been identified as potential therapeutic targets for MODS.

Histones are highly cationic intra-nuclear proteins that support the normal structural development of chromatin and regulation of gene expression. Histones and DNA-bound histones (nucleosomes) are released into the extracellular space during cell death processes including necrosis, apoptosis and neutrophil extracellular trap-induced cell death (NETosis). In the extracellular space, histones act as cytotoxic damage-associated molecular pattern proteins by activating Toll-like receptors (TLRs), promoting pro-inflammatory cytokine pathways and altering phospholipid membrane permeability[3,7–9]. In humans, the normal serum level of histones is very low (estimated at <0.6 ng mL$^{-1}$)[10–12]. However, serum levels as high as 3 ng mL$^{-1}$ have been reported in critically ill patients, and correlate with hallmark features of MODS including, coagulopathy, endothelial dysfunction and inflammation[13–16].

Several animal studies demonstrate that intravenous administration of exogenous histones is sufficient to cause a MODS-like phenotype[3,7,17]. Importantly, anti-histone treatments (e.g., histone neutralizing antibodies, activated protein C (APC), recombinant thrombomodulin and heparin) protect mice against secondary organ failure due to lethal endotoxemia, sepsis, ischemia/reperfusion injury, trauma, pancreatitis, peritonitis, stroke and thrombosis[2–4,18,19]. However, therapeutic approaches currently pursued in experimental models have marked limitations. For example, despite their use in laboratory experiments, TLR2/4 monoclonal antibodies (mAbs) to block extracellular histone signaling would cause substantial immunodeficiency in humans by inhibiting innate immune defenses after host infection. Similarly, anti-histone mAbs have been implicated in autoimmunity[20,21]. Several other biologics that have demonstrated efficacy in animal models failed to provide therapeutic benefit in clinical trials (e.g., APC) and are associated with increased risk of bleeding (e.g., heparin and APC) or systemic toxicity (e.g., histone deacetylase inhibitors)[22,23]. In addition, many biologics require restrictive handling and storage, special dosing considerations and risk allergic reactions (recombinant proteins and antibodies), which limit their use in field situations or small regional health clinics[24,25]. Thus, the development of specific histone inhibitors is a unique clinical opportunity to interrupt a pathophysiologic cascade responsible for significant morbidity and mortality associated with MODS.

Chemically stabilized nucleic acid bio-drugs (aptamers) are synthetic structure RNA or DNA oligonucleotide ligands that bind with high affinity and specificity to their targets[26–28]. As a therapeutic, aptamers possess several key advantages over other biologics, including the following. (1) They are self-refolding, single chain and redox insensitive. Unlike proteins, aptamers tolerate pH and temperatures that proteins do not. (2) They act as selective inhibitors of their target, eliminating potential for off-target effects. (3) They are easy and economical to produce. Their production does not depend on bacteria, cell cultures or animals. (4) Their small size leads to a high number of moles of target substance bound per gram of aptamer. Additionally, their transport properties allow for improved tissue penetration. (5) They are stable at ambient temperature, yielding a much longer shelf-life than other biologics, and can tolerate transportation without refrigeration. (6) Cross-species reactive aptamers can be easily engineered, thus expediting testing of the same reagent in preclinical animal models and in future human clinical trials. The clinical potential of aptamers is also highlighted by the Food and Drug Administration (FDA) approval of an aptamer-based drug for the treatment of macular degeneration and by clinical trials demonstrating the safety and efficacy of systemically administered aptamers[29–35].

In this study, we develop an anti-histone therapeutic strategy to selectively neutralize extracellular histones implicated in MODS, based on aptamers. Because histones (cationic proteins) normally associate with DNA in the nucleosome, oligonucleotides such as aptamers (anionic molecules) have intrinsic high affinity for histones, making them an excellent reagent for binding and neutralizing extracellular histones and reducing the morbidity and mortality associated with MODS. In this study, we develop nuclease-resistant, 2′ fluoro-modified RNA aptamers with high affinity ($K_D$ = nanomolar (nM)) for histones implicated in MODS (H3 and H4) with systematic evolution of ligands by exponential enrichment (SELEX)[36,37]. A key stringent negative selection step was introduced during the selection process to favor the identification of aptamers that bound to histones, but not other proteins in human serum, thereby reducing potential side effects. We show that the aptamers inhibit histone-induced human platelet aggregation and endothelial cell death in culture. In addition, we demonstrate efficacy of anti-histone aptamers in a murine model of MODS. Because extracellular histones have been implicated in the development of many different disease conditions, the anti-histone aptamer bio-drugs could potentially impact the treatment of numerous MODS-inducing clinical conditions.

## Results

**Identification of histone-specific RNA aptamers using SELEX.** Two parallel selections were performed using the in vitro SELEX protocol to isolate RNA aptamers (51 nucleotides in length) that selectively bind to human histones implicated in MODS (H3 and H4) (Fig. 1a). To identify RNA sequences that specifically bind to histones, but not to proteins in serum, we introduced a stringent negative selection step against bovine serum albumin (BSA) and human IgG (Fig. 1a; step 2). RNAs that bound to serum proteins were removed (Fig. 1a; step 3) and unbound RNA was recovered. To ensure the isolation of high-affinity binding RNAs to human histones, a positive selection step was performed using human recombinant histones H3 and H4 proteins (Fig. 1a; step 3). Eight rounds of selection (against target histones H3 or H4) and negative selection (against BSA and human IgG) were performed (see Supplementary Table 1). Binding of the round 0 (R0) and round 8 (R8) RNA pools to histones H3 and H4 was verified using the double-filter binding assay (Fig. 1b; top panels). A leftward shift in the binding curve of the R8 RNA pools compared to the R0 RNA pools for both selections is indicative of enrichment of high-affinity binding RNA sequences. Importantly, the enriched R8 RNA pools did not bind to serum proteins (Fig. 1b; bottom panels) confirming specificity for histones. Equilibrium dissociation constants ($K_D$) for R0 and R8 RNA pools were determined using one site binding nonlinear regression curve fit (see Supplementary Table 2).

High-throughput sequencing (Illumina) and bioinformatics analysis of selected RNA sequences from the selection rounds

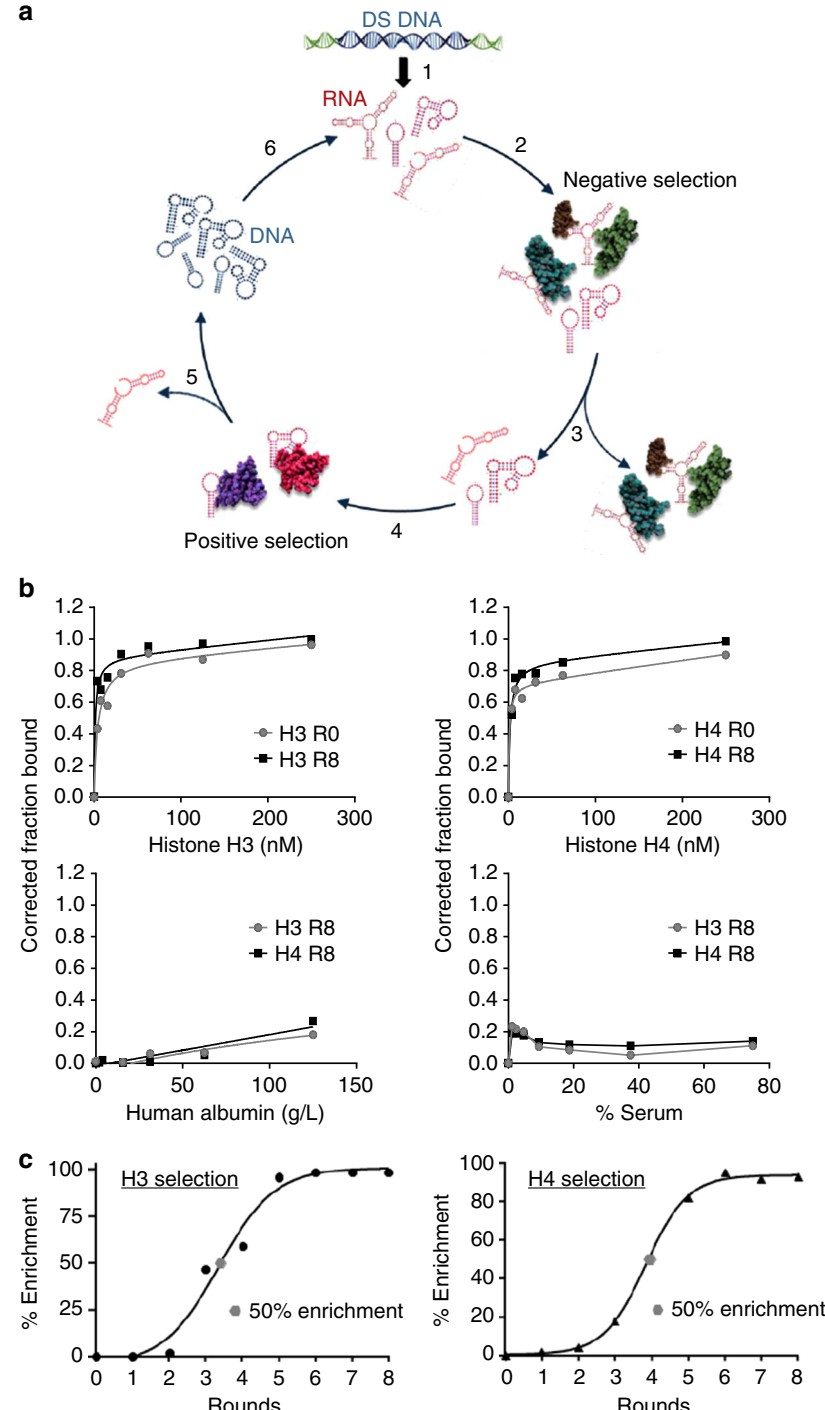

**Fig. 1** Identification of histone-specific RNA aptamers using SELEX. **a** Schematic of the in vitro Systematic Evolution of Ligands by Exponential Enrichment (SELEX) procedure. Step 1. Double-stranded DNA (DS DNA) template library (Sel2N20) is in vitro transcribed in the presence of 2′ Fluoro pyrimidines and 2′ OH purines to generate the 2′ Fluoro-modified Round 0 RNA library (RNA). Step 2. The round 0 RNA library was incubated with human albumin and human IgG to remove RNAs that bind to human serum proteins (Negative selection). Step 3. RNA bound to serum proteins was discarded. Step 4. RNA not bound to serum proteins was incubated with human histones H3 and H4, respectively. Step 5. Histone-bound aptamers were collected and reverse-transcribed into DNA. Step 6. Round 1 DNA was then transcribed into RNA for the subsequent round of selection. A total of eight rounds of selection were performed for each histone selection (see Supplementary Table 1). **b** Binding of Round 0 (R0) and Round 8 (R8) RNA to recombinant human histone H3 (top, left panel) and H4 (top, right panel) proteins. Binding of R8 RNA to human albumin (bottom, left panel) and human serum (bottom, right panel). **c** Percent sequence enrichment (% Enrichment) at each round of selection (black circle). The 50% sequence enrichment point (gray circle) is indicated for each selection

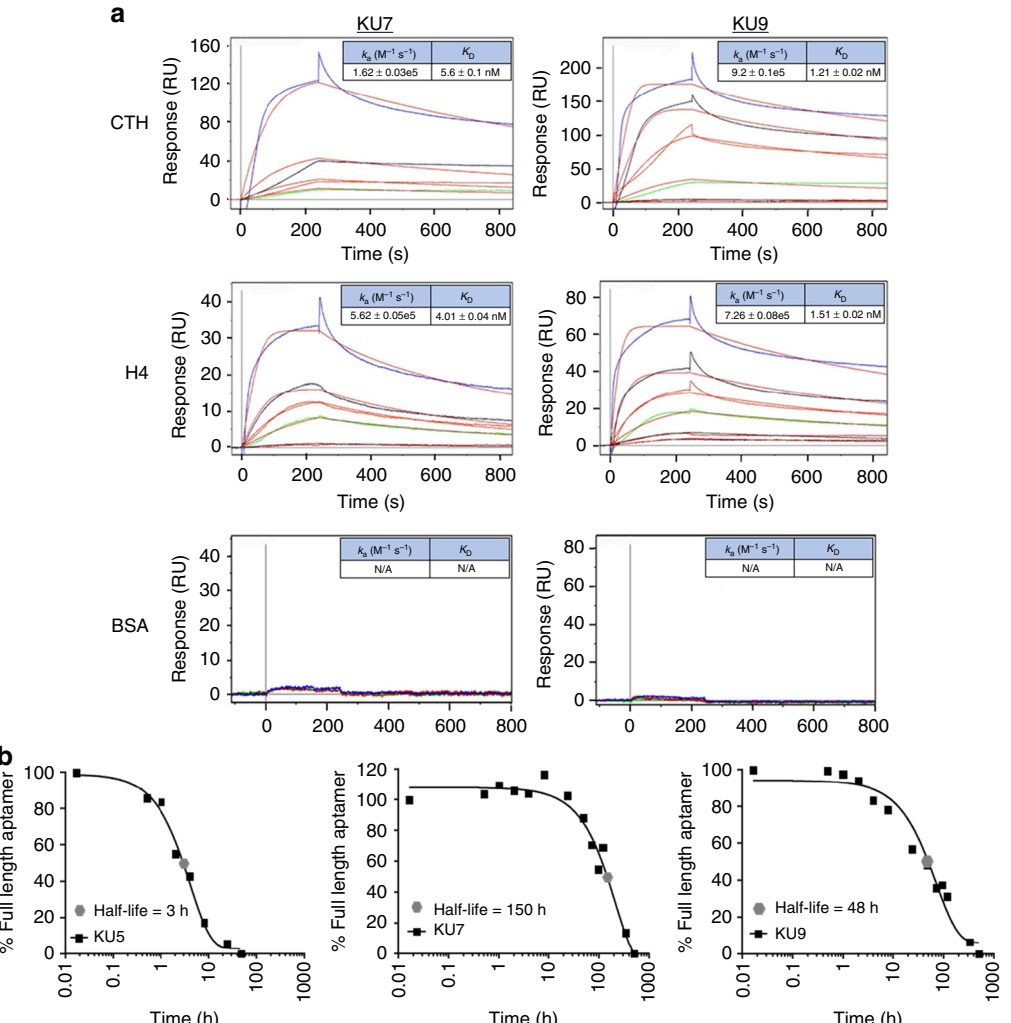

**Fig. 2** Binding characterization and stability measurements of individual histone RNA aptamer sequences. **a** Binding kinetic rate constants ($k_a$ and $K_D$) determined for aptamers KU7 (left panels) and KU9 (right panels) binding to CTH (top panels), H4 (middle panels) and BSA (bottom panels). Aptamers concentrations tested: 100 nM (blue), 50 nM (black), 25 nM (red), 12.5 nM (green), 10 nM (magenta). **b** Serum stability measurements for aptamers KU7 and KU9 (5 µM) in 50% human serum. $T_{1/2}$ KU7 = 150 h. $T_{1/2}$ KU9 = 48 h

revealed that the selections converged between rounds 5 and 8 (Fig. 1c). Percent sequence enrichment (% enrichment) was measured to determine the evolution of each sequence family at the later rounds of selection. Percent sequence enrichment at each round of selection was measured by taking the total number of unique sequences in each round and dividing by the total number of sequences obtained in each round. For each selection, the 50% enrichment point is indicated by a gray dot (Fig. 1c). The top three RNA sequences from each selection (H3 selection: KU4–KU6; H4 selection: KU7–KU9) were selected based on the following criteria: sequences that increase in abundance during the positive selection rounds (against target histones H3 or H4) but do not decrease in abundance during the negative selection rounds (against BSA and human IgG) (see Supplementary Table 1 for details on selection conditions). Individual RNA sequences are listed in Supplementary Table 3. Theoretical secondary structures for each of the six RNA aptamers were generated using Mfold[38] (Supplementary Fig. 1).

Next, the binding of single RNA aptamers isolated from the histone H3 and H4 selections was evaluated with surface plasmon resonance (SPR) (Fig. 2a) and double-filter binding assay (Supplementary Fig. 2; data shown for aptamers KU7 and KU9). RNA aptamers were incubated with either calf thymus-

derived histones (CTH) (top panels), recombinant human histone H4 (middle panels) or BSA (bottom panels) (Fig. 2a). As predicted, the RNA aptamers bound with high affinity to both CTH histones and recombinant human histone H4, but not to BSA (Fig. 2a). Importantly, the aptamers bound equally well to both CTH and recombinant human histone H4 suggesting that the aptamers bind irrespective of post-translational modifications and are not species specific. While aptamers KU7 and KU9 were derived from the H3 selection, we also evaluated binding of the RNAs to all histone subunits using the double-filter binding assay (Supplementary Fig. 2A). Of note, both aptamers bound with low nM affinity to histones, H1, H3 and H4 (major histones implicated in MODS). In contrast, while only aptamer KU7 bound to histone H2B, neither aptamer bound to histone H2A. In addition, equilibrium binding kinetics were determined for one of the aptamers (KU7) using the double-filter binding assay. We observed similar (low nM) binding affinities at various time intervals (Supplementary Fig. 2B). Finally, as observed with SPR analysis (Fig. 2), no significant binding was observed to human serum or human albumin (Supplementary Fig. 2C), confirming specificity of the aptamers for histone proteins vs. serum proteins. Of note, differences in the binding constants generated using the double-filter binding and SPR assays could be due to

inherent differences with the assays and binding buffers used. For example, the double-filter binding assay is performed in solution (e.g., histone and aptamers are free to interact in solution). In contrast, binding with SPR is assessed by immobilizing the ligand (e.g., histones) on a solid surface. Together, these data highlight the potential safety of this approach for targeting extracellular histones, while minimizing unwanted off-target effects.

Based on the predicted secondary structures of the RNA aptamers (see Supplementary Fig. 1), we reasoned that two of the aptamers (KU7 and KU9) would be more stable in human serum compared to the other four aptamers (KU4, KU5, KU6 and KU8) and thus better suited for clinical applications. To assess their serum stability, RNA aptamers were incubated with 50% human serum for up to 28 days. Aliquots of the aptamer-serum solution, obtained at various time points, were resolved on a denaturing polyacrylamide gel electrophoresis (PAGE) gel (8 M urea 10% acrylamide) following staining with SYBR gold to visualize the RNAs (Supplementary Fig. 3). The amount of full-length aptamer was quantified for each time point by comparing the intensity of the bands to the "0 h" time point using the Image Lab software. (Fig. 2b; data shown for aptamers KU5, KU7 and KU9). As predicted, aptamers KU7 and KU9 had superior serum stability profiles compared to the other aptamers tested (comparison shown for KU5 only).

**In vitro efficacy of RNA aptamers**. The release of histones from dying cells is associated with microvascular thrombosis and tissue ischemia[4,39]. Histone H4 and, to a lesser extent H3, are responsible for directly inducing aggregation of human platelets. To determine whether the selected RNA aptamers inhibit adverse effects of extracellular histones, aptamers KU7 and KU9 were incubated with platelets derived from healthy human donors in the absence or presence of histone H4. We show that anti-histone aptamers KU7 and KU9 inhibit histone H4-induced platelet aggregation in a dose-dependent manner (Fig. 3a). In contrast, the aptamers did not inhibit platelet aggregation induced by collagen, even at the highest aptamer dose tested (Fig. 3a). Importantly, histone-induced platelet aggregation was significantly inhibited by aptamers KU7 and KU9 at a molar ratio of histone to aptamer of 1:1, 2:1 and, to a lesser extent, 4:1 (Fig. 3a). Platelet aggregometer graphs for the individual aptamers and RNA round pools are shown in Supplementary Fig. 4A and B.

In vivo, the release of extracellular histones in circulation results in an amplifying effect by promoting additional cell death and nuclear content release, as well as activation of TLRs resulting in pro-inflammatory cytokine production (interleukin-6 (IL-6)), enhanced inflammation and coagulation activation[7,13–15]. We show that aptamers KU7 and KU9 inhibit CTH-induced TLR activation as measured by IL-6 production (Fig. 3b and Supplementary Fig. 4C; left panel). Importantly, histone-induced TLR activation was significantly inhibited by aptamers KU7 and KU9 at a molar ratio of histone to aptamer of 1:1, 2:1, 4:1 (Fig. 3b). In contrast, the aptamers have no effect on lipopolysaccharide-induced TLR activation (Supplementary Fig. 4C; right panel).

In addition to a heightened inflammatory response, high levels of extracellular histones are cytotoxic to endothelial and epithelial cells as well as several other cell types[2,19,40,41]. We confirmed that administration of calf thymus histones to a human hybrid endothelial cell line (EA.hy926) causes dose-dependent cell death (Fig. 3c; left panel). Aptamers (KU7 and KU9) have a dose-dependent protective effect in neutralizing histone-induced cytotoxicity (Fig. 3c; right panel).

Extracellular histones appear to induce cytotoxicity through their interaction with phospholipids in the plasma membrane, leading to transmembrane conductance, calcium influx, cell swelling and cytolysis[42–44]. We confirmed that administration of calf thymus histones to EA.hy926 cells resulted in increased intracellular calcium [$Ca^{2+}$] levels[+]) calcium influx (Fig. 3d). Importantly, aptamers significantly inhibit the histone-induced increase of intracellular calcium at molar ratio of histone to aptamer up to 4:1 with KU7 and 6:1 with KU9 as measured using the calcium indicator Fura 2-acetoxymethyl ester (Fura 2-AM). The aptamers alone had no effect on intracellular calcium levels (Fig. 3d). Together, these data confirm that the aptamers can prevent the functional effect of histones in vitro and provide the rationale for proposing that these aptamers have the potential to attenuate histone-mediated injury in vivo.

**Efficacy of histone aptamer in a murine model of MODS**. Histone administration causes neutrophil migration, endothelial injury and dysfunction, hemorrhage and thrombosis and often results in death at levels as low as 50 mg kg$^{-1}$ body weight. Intravenous injection of histones into mice leads to organ dysfunction and death (Fig. 4). Death occurred between 1 and 3 h in the mice treated with histones alone, but was inhibited by administration of aptamer (Fig. 4a). Injection of histones resulted in an increase in organ weight that was only partially reversed by the aptamer treatment (Fig. 4b). The increase in organ weight is likely a result of an increase in vascular congestion, multifocal neutrophilic aggregates in vessels, thrombi and hemorrhage (Fig. 4c, d; the liver, lung and spleen shown in Supplementary Fig. 5). Importantly, while no significant change in organ weight was observed with aptamer treatment, pathologic evidence of histone toxicity, as determined by hematoxylin and eosin (H&E) staining and immunostaining for presence of platelets and neutrophils in the various tissues, was inhibited by aptamer KU7 at a ratio of histone to aptamer of 2:1 and 4:1 (Fig. 4c, d). As predicted, injection of histones into the circulation of mice also resulted in an increase in TLR activation via IL-6 (Fig. 4e). The increase in TLR activation was partially attenuated by aptamer treatment in the liver but not in lung or spleen, and is likely due to the retention of aptamer–histone complexes in these organs. Together, these data confirm that aptamers that bind to the histones implicated in MODS can attenuate histone-mediated injury in vivo.

The experimental protocol of the in vivo efficacy study described above (Fig. 4) is consistent with all previous studies performed in preclinical models of MODS with histone inhibitors[16,45,46], such that the therapeutic treatment is administered prior to, or in conjunction with, the histones. To better represent the clinical scenario where the drug is administered after tissue injury, we evaluated the ability of anti-histone aptamers to rescue histone-induced toxicity as a function of time (Fig. 5). Anti-histone aptamers were added to EA.hy926 cells in culture at different time points following the addition of histones. Anti-histone aptamers prevented cytotoxicity in a time-dependent manner when administered after the histones, demonstrating the ability to protect from histone-induced cell death 2 h after histone administration (Fig. 5a). The administration of KU7 rescued the increase in intracellular [$Ca^{2+}$] after histones had already induced an increase of [$Ca^{2+}$] (Fig. 5b). As shown in Fig. 4a, intravenous injection of histones resulted in death of mice within 3 h; however, mice treated with anti-histone aptamer KU9 at 30 min following histone injection had markedly improved survival as compared to vehicle-treated mice (Fig. 5c; left panel) and a reduction in lung weight (Fig. 5c; right panel).

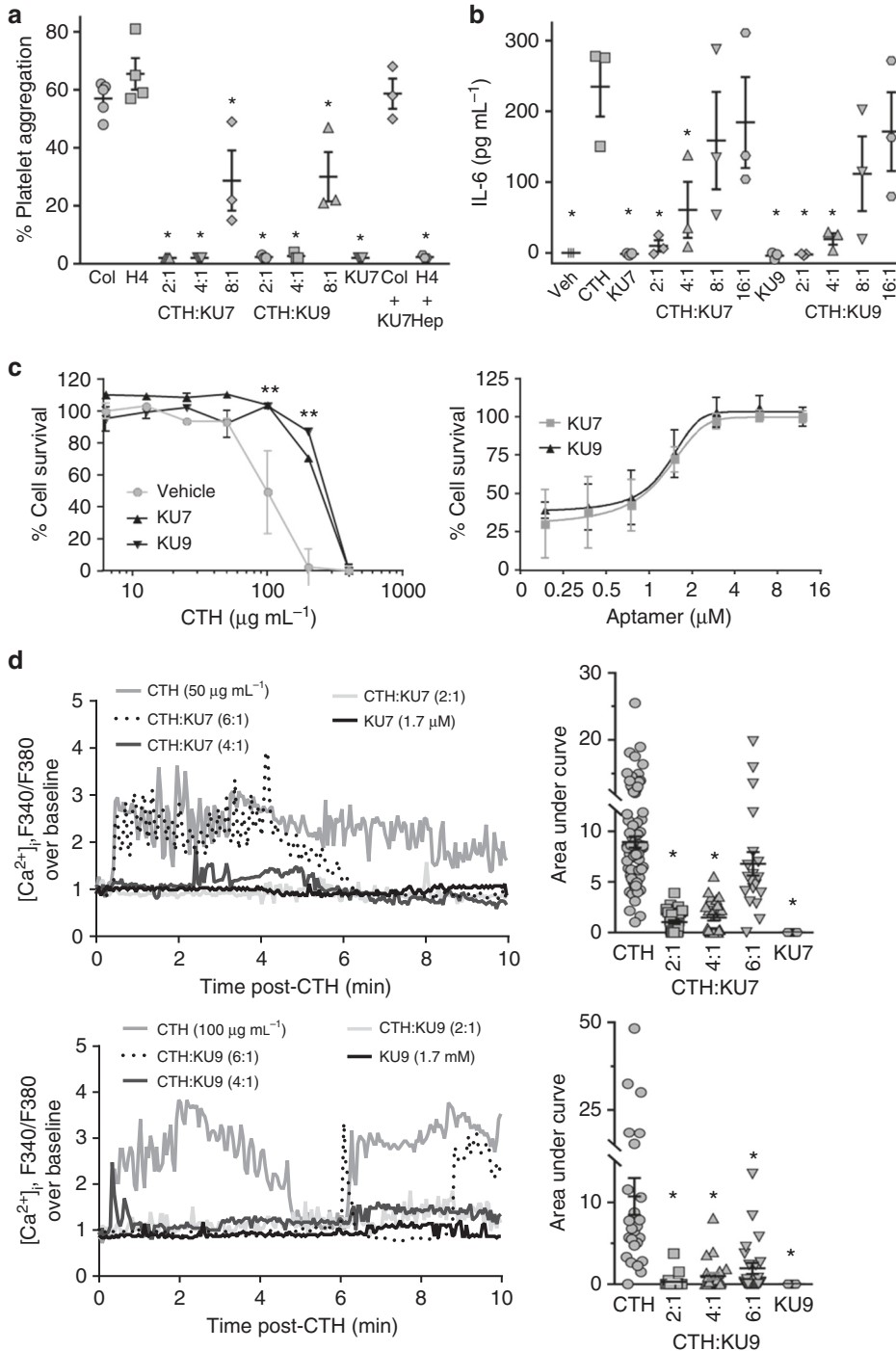

**Fig. 3** In vitro efficacy of RNA aptamers. **a** Human platelet aggregation measurements using platelets derived from three independent healthy donors. Collagen (Col), histone H4 (H4), histone aptamers (KU7 and KU9), calf thymus histones (CTH), heparin (Hep); *p < 0.01 vs. H4, one-way ANOVA corrected for multiple comparisons (Dunnett testing), n = 3–5. **b** TLR activation: IL-6 ELISA detection of IL-6 protein levels (as a measurement of TLR activation) in supernatants of EA.Hy926 cells treated with media alone (Veh), calf thymus histones (CTH), histone aptamers (KU7 and KU9) alone or in combination; *p < 0.01 vs. CTH, one-way ANOVA corrected for multiple comparisons (Dunnett testing), n = 3. **c** Cytotoxicity measurements: aptamer inhibition of histone-mediated cytotoxicity of endothelial cells determined by MTS assay. (Left panel) EA.hy926 cells treated with 1.2 μM of aptamers (KU7 or KU9) and varying amounts (0 to 1000 μg mL⁻¹) of calf thymus histones (CTH). (Right panel) EA.hy926 cells treated with 180 μg mL⁻¹ of CTH and increasing amounts (0 to 16 μM) of aptamers (KU7 or KU9); **p < 0.01 vs. vehicle (100 μg mL⁻¹), n = 4. **d** Dynamic changes of intracellular calcium levels in Fura 2-AM-loaded EA.hy926 cells using fluorescence microscopy. (Left panel) Representative intracellular calcium elevation traces (F340/F380) of EA.hy926 treated with CTH alone, aptamer KU7 alone (25 μg mL⁻¹, top panels), aptamer KU9 alone (25 μg mL⁻¹, bottom panels) or in the presence of varying aptamer amounts (Molar ratio of CTH to aptamer indicated). (Right panel) Summary of data from multiple EA.hy926 cells (n = 19–69 cells per bar); **p < 0.01 vs. CTH, one-way ANOVA corrected for multiple comparisons (Tukey's testing), n = 3. Data represent mean ± SEM

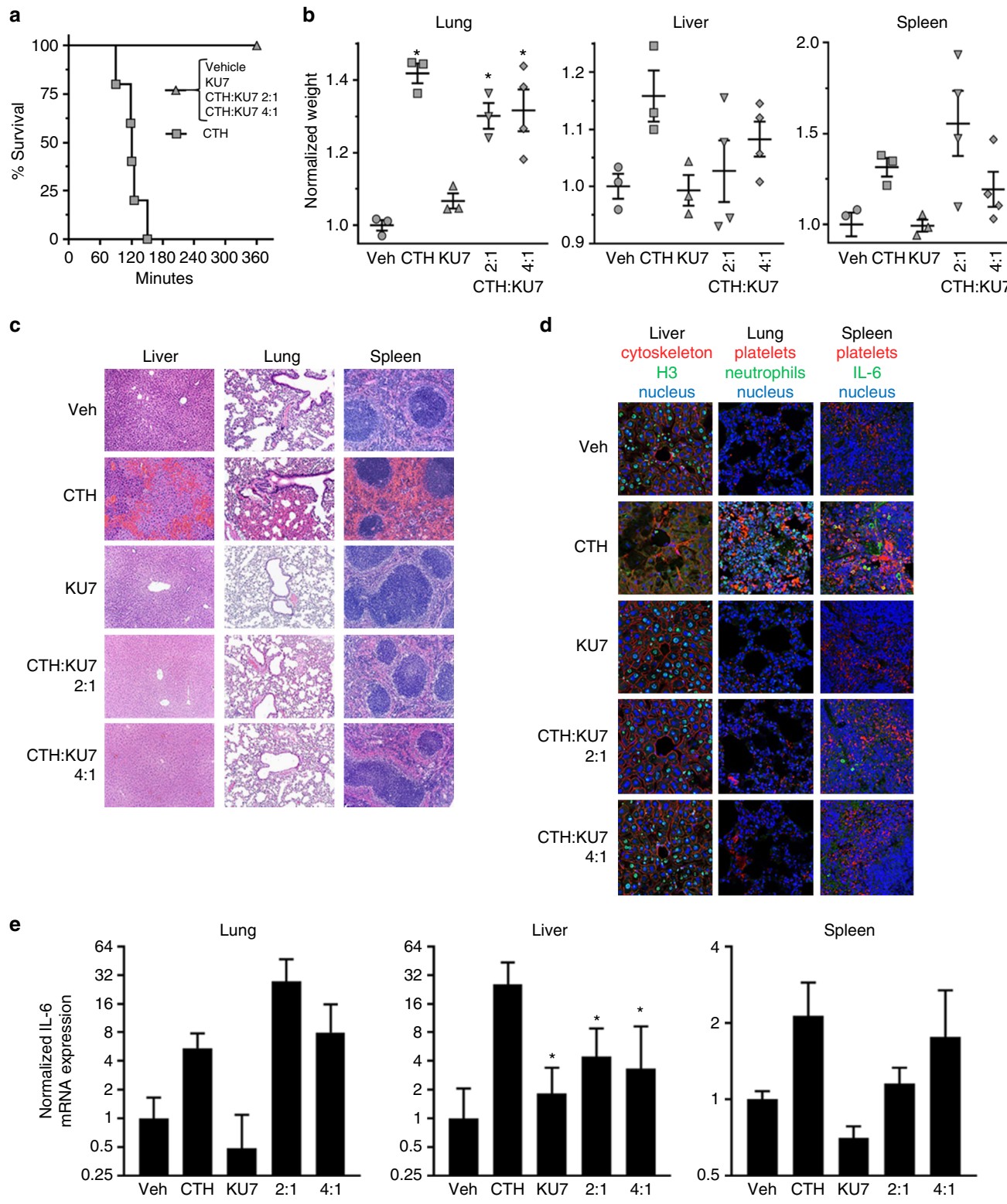

**Fig. 4** Efficacy of histone aptamer in murine model of MODS. **a** Survival curves of mice injected IV with CTH in the presence or absence of aptamer treatment; $n = 6$ per group. Molar ratio of CTH to aptamer indicated. **b** Weights of the liver, lung and spleen normalized to pre-treatment body weight; *$p < 0.05$ vs. Veh, one-way ANOVA corrected for multiple comparisons (Dunnett testing), $n = 4$–6 per group, data represent mean ± SEM. **c** Histology of the liver, lung and spleen of vehicle, CTH, KU7, CTH+KU7 (2:1), and CTH + KU7 (4:1) treated mice. Vascular congestion, thrombi and hemorrhaging are observed in the CTH mice. Scale bar = 100 μm. **d** Immunostaining of mouse liver, lung and spleen stained to identify nucleus (pseudo blue), cytoskeleton and platelets (pseudo red), histone H3, neutrophils and IL-6 (pseudo green) (see Supplementary Fig. 4 for quantitative analysis of immunostaining data). Scale bar = 25 μm. **e** Normalized IL-6 mRNA expression levels in the liver, lung and spleen of mice. Normalized IL-6 mRNA expression was determined using the following equation: $2^{(\Delta-\Delta CT)}$. $\Delta CT$ = beta actin − IL-6, $\Delta-\Delta CT$ = Vehicle $\Delta CT$ − Treatment $\Delta CT$. Error bars represent the lower limit (LL) and upper limit (UL) of the standard deviation from the $\Delta-\Delta CT$ [$2^{(LL\ \Delta-\Delta CT)}$ and $2^{(UL\ \Delta-\Delta CT)}$]; *$p < 0.05$ vs. CTH (50 mg kg$^{-1}$)

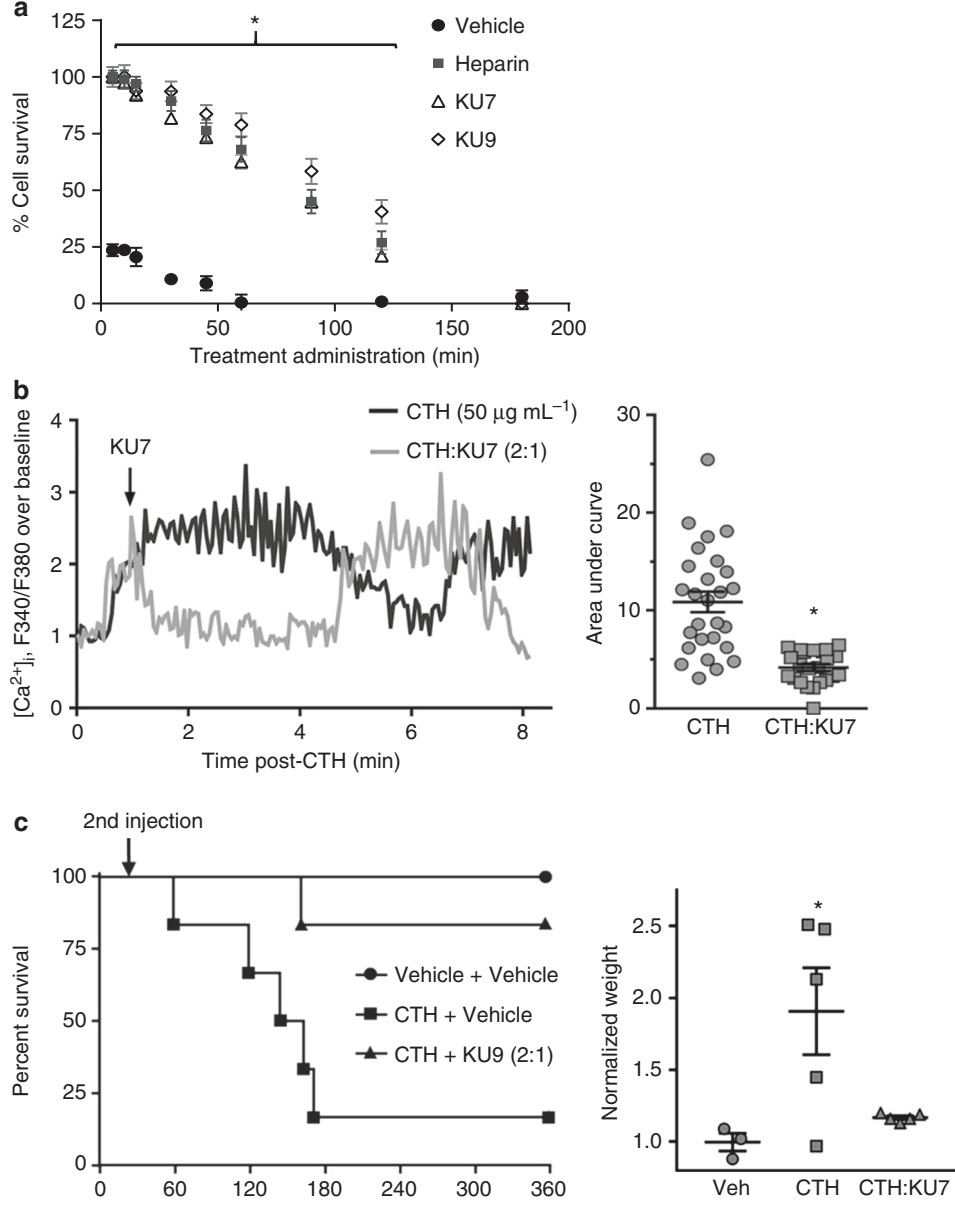

**Fig. 5** Efficacy of anti-histone aptamers when administered after histones. **a** Aptamer inhibition of histone-mediated cytotoxicity of endothelial cells determined by MTS assay. EA.hy926 cells treated with 200 μg mL$^{-1}$ of calf thymus histones followed by administration of either vehicle (negative control), heparin (positive control, 1:1), KU7 aptamer (1:2) or KU9 aptamer (1:2) at time points of 0, 5, 10, 15, 30, 45, 60, 90, 120 and 180 min after CTH; $n = 3$ biological replicates; *$p < 0.0025$ vs. vehicle. **b** Dynamic changes of intracellular calcium levels in Fura 2-AM-loaded EA.hy926 cells using fluorescence microscopy. (Left panel) Representative intracellular calcium elevation traces (F340/F380) of EA.hy926 treated with CTH (50 μg mL$^{-1}$) then followed by addition of vehicle or aptamer KU7 (Molar ratio of CTH to aptamer 2:1) added 1 min after CTH. (Right panel) Summary of data (*$p < 0.0001$ vs. CTH, unpaired two-tailed $t$-test, $n = 23-27$ cells). **c** Efficacy of anti-histone aptamers in murine model of MODS. Survival curves of mice injected IV with vehicle, or CTH (62.5 mg kg$^{-1}$), followed in 30 min by IV injection of vehicle or aptamer KU9 (31.25 mg kg$^{-1}$, indicated as 2nd injection); $n = 6$ biological replicates; *$p < 0.05$ vs. CTH+vehicle. Lung weight normalized to pre-treatment body weight. *$p < 0.05$ vs. Veh, one-way ANOVA corrected for multiple comparisons (Dunnett testing), $n = 3-5$ per group, data represent mean ± SEM

**Aptamers bind and neutralize NET toxicity**. Recent data demonstrate the importance of NETs as an important source of extracellular histones in a variety of clinical disease including MODS, sepsis, rheumatologic diseases and thrombosis[16,47–49]. We therefore assessed whether our anti-histone aptamers could prevent NET-associated cell toxicity. Human neutrophils were isolated, stimulated with phorbol myristate acetate (PMA) and NETs were generated for immunofluorescence microscopy and cell toxicity assays (Fig. 6). The 4′,6-diamidino-2-phenylindole

(DAPI) stain demonstrates the formation of extracellular NETs (Fig. 6a; top, left panel) displaying histones (Fig. 6a; top, right panel). As shown in Fig. 6a, aptamer KU7 binds to human neutrophil-derived NETs (Fig. 6a, bottom, right panel). White color areas in the merged images demonstrate the close approximation of the aptamer KU7 with NETs (Fig. 6a; bottom, right panel). Importantly, anti-histone aptamer KU7 also attenuated NET-induced cell death (Fig. 6b), confirming their potential as a viable therapeutic option for clinical conditions (e.g., sepsis)

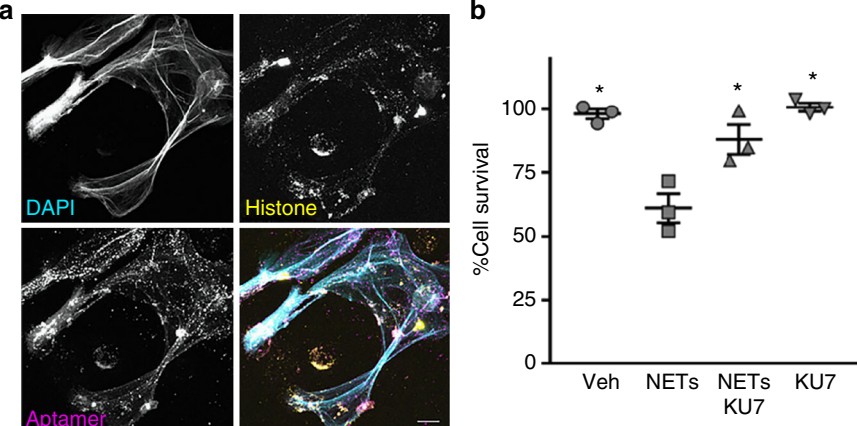

**Fig. 6** Aptamers bind to human neutrophil-derived NETs and inhibit NET-induced cytotoxicity. **a** Confocal microscopy of human neutrophil-derived NETs. Single images are shown in gray scale. DAPI (top, left); histones (top, right); aptamer KU7 (bottom, left); merged images show: cyan, DAPI labeling of DNA, yellow, histones, and magenta, aptamer KU7–647 (bottom, right). White areas in the merged image represent close proximity of DNA, histone and aptamer. Representative images, captured with 40× oil and 2.8× zoom. Scale is equivalent to 10 μm. **b** Aptamer inhibition of NETs-mediated cytotoxicity of endothelial cells determined by MTS assay. EA.hy926 cells treated with 8 μg per well of NETs material (based on DNA concentration) and/or 8 μg per well (10.66 μM) of aptamer (KU7); *$p < 0.01$ vs. NETs, one-way ANOVA corrected for multiple comparisons (Tukey), $n = 3$, data represent mean ± SEM

leading to MODS. Although previous reports of histone inhibitors have demonstrated efficacy in the MODS via simultaneous administration of anti-histone therapy, this is the first report of an anti-histone therapy capable of rescuing the damaging cellular effects of extracellular histones in vitro and in vivo.

## Discussion

This is the first report of an aptamer-based therapeutic approach for MODS. In this study, we developed nuclease-resistant RNA aptamers that bind with high affinity and specificity to human histones H3 and H4 implicated in MODS. Importantly, the aptamers do not bind to other proteins in human serum, and thus are expected to be safe for use in humans. We show that the aptamers inhibit histone-induced platelet aggregation (Fig. 3a), TLR activation (Fig. 3b) and endothelial cell death (Fig. 3c) and calcium influx (Fig. 3d) in human cells in culture. Importantly, these aptamers bind human NETs and neutralize their toxicity in vitro (Fig. 6). Finally, we demonstrate efficacy of one of the anti-histone aptamers (KU7) in a murine model of MODS (Figs. 4 and 5).

A key consideration when developing a drug that targets histones directly is that the drug must act exclusively in the extracellular space. Indeed, histone neutralizing drugs that obtain intracellular access could potentially disrupt DNA structure or function, and result in catastrophic side effects. To this end, oligonucleotide aptamers are highly anionic, and thus membrane impermeable which would negate any concerns with regards to toxicity due to targeting chromatin. An additional safety feature engineered into our anti-histone aptamers was the minimization of potential interactions between the aptamers and serum proteins. This was achieved by performing a stringent negative selection step (against human IgG and BSA) at every round of selection. An additional benefit imparted by the negative selection step was the maximization of the fraction of the aptamer drug that binds histones in circulation, thereby reducing the therapeutic dose of the aptamer. While binding of oligonucleotide-based drugs to serum proteins is normally viewed as a benefit to improve the pharmacokinetic properties of the drug and prolong circulating times, previous therapeutic aptamers to specific serum proteins (e.g., thrombin and factor IX) which have shown efficacy in humans were developed to minimize off-target interactions

with non-target proteins in serum[26,50,51]. In addition, binding of the histone aptamers to multiple histone subtypes could also have a therapeutic advantage by enabling the inactivation of several histone proteins implicated in MODS with one aptamer drug. Similarly, a pool of (unselected) 2′ F-modified RNA aptamers, with low affinity for serum proteins, could be used to neutralize all cationic nuclear proteins (including all histones subtypes) released in circulation following extensive cell injury, thereby maximizing therapeutic efficacy.

Three distinct therapeutic approaches have been described to target extracellular histones. These include pharmacological agents to block histone release (e.g., PAD4 inhibitors and DNAse 1)[52,53] or downstream signaling (e.g., TLR blocking mAb or C reactive protein) or agents that directly neutralize extracellular histones (e.g., anti-histone mAb, APC and heparin)[7,19,48,54,55]. Although these approaches have shown promise in animal models of MODS, further research is necessary to warrant their safe application in a clinical setting. Indeed, one of the common problems with many of these anti-histone approaches is their propensity for nonspecific, off-target effects resulting in systemic toxicity.

Oligonucleotide-based drugs have been under clinical development for the past 30 years, starting with anti-sense oligos and followed by aptamers and small interfering RNAs. One of the key factors contributing to the approval of six oligonucleotide drugs over recent years has been the advent of nucleic acid chemistry to improve the safety and stability of these drugs. For example, the anti-vascular endothelial growth factor (VEGF$_{165}$) aptamer drug Macugen (formerly pegaptanib), which was approved by the FDA in 2004 to treat age-related macular degeneration of the retina, contains a phosphorothioate 3′−3′ deoxythymidine cap to promote nuclease stability, 2′-O-methylated purines and 2′-fluorinated pyrimidines. Finally, a 40 kDa molecular weight polyethylene glycol substituent was linked to the 5′ terminus of the aptamer to increase retention time in the eye[56,57]. Another safety consideration when developing the pegaptanib aptamer was enhancing target specificity. The pegaptanib aptamer was selected to be highly specific for VEGF$_{165}$, the predominant VEGF isoform in the human eye, thereby reducing potential systemic effects and greatly boosting the safety profile of the drug. The anti-histone aptamers developed in this study, which are modified with 2′-fluoro chemistry, were shown to be extremely

stable in human serum ($T_{1/2}$ KU7 = 150 h and $T_{1/2}$ KU9 = 48 h) (Fig. 2b) and safe in vivo (Fig. 4). In addition, as discussed above, the anti-histone aptamers were engineered to selectively bind histones in serum.

One additional benefit of the modification of the 2′-position of the ribose is increased duplex stability ($T_m$). 2′F modification in particular polarize the bases more significantly and contribute to strengthening of Watson-Crick base pairing[58]. For example, in the RNA interference field the combined use of 2′-fluoro pyrimidines with 2′-O-methyl purines results in RNA duplexes with extreme stability in serum and improved in vivo performance. Chemically engineered oligonucleotides with 2′-chemistry have also been used as efficient and specific silencers of endogenous microRNAs (miRNAs) in mice[59,60]. This is attributed to the ability of these modified oligos to compete with endogenous miRNAs for binding to their target mRNAs[60]. Similarly, post-selection modification of the 2′-position of aptamers increases binding to the target[61–63] and possibly favors binding of the anti-histone aptamers to both free histones and DNA-bound histones (nucleosomes) in serum (also see binding of aptamers to human neutrophil-derived NETs, Fig. 6a). Since free histones and DNA-bound histones are both pro-inflammatory[9], a single drug capable of neutralizing both forms is optimal.

In summary, we describe a approach using nuclease stabilized RNA aptamers to neutralize the toxic effects of extracellular histones implicated in MODS. Our approach overcomes many of the limitations of other experimental anti-histone treatments and provides a promising avenue for clinical implementation of a robust therapeutic for MODS in combination with other interventions (anti-inflammatory and supportive care). Because extracellular histones have been implicated in the development of many different disease conditions, the anti-histone aptamer biodrugs could potentially impact the treatment of numerous MODS-inducing clinical conditions including acute lung injury, transfusion-related acute lung injury, sepsis, trauma, burns, stroke, cancer, autoimmune/inflammatory disorders and ischemia/reperfusion and drug-mediated tissue injury.

## Methods

**Cell lines and cell culture reagents**. Ea.Hy926 cells (ATCC®, CRL-2922, Manassas, VA) were maintained in Dulbecco's modified Eagle's medium (GIBCO, Gaithersburg, MD) supplemented with 10% fetal bovine serum (Atlanta Biologicals, Flowery Branch, GA) at 37 °C under 5% CO2. The medium was changed every 2 to 3 days until confluent. During passaging, cells were trypsinized with Trypsin-EDTA (0.25%) (GIBCO, Gaithersburg, MD) and re-plated at a 1 to 4 ratio.

**Selection protocol**. The protocol for the generation of the initial (round 0) RNA Library has been described elsewhere. Step 1: Generation of double-stranded DNA (dsDNA) template by elongation. Elongation Solution 1 components: 9 μL of 100 mM Tris-HCl, pH 8.0 (Sigma-Aldrich, St. Louis, MO), 15 μL of 50 mM MgCl2 (Sigma-Aldrich, St. Louis, MO), 1000 pmol of Sel2N20 template oligo (5′-TCGGGCGAGTCGTCTG-N20-CCGCATCGTCCTCCC-3′) (IDT, Coralville, IA), 2000 pmol Sel2 5′ primer (5′-TAATACGACTCACTA-TAGGGAGGACGATGCGG-3′) (IDT, Coralville, IA) and UltraPure distilled H2O (diH2O) (ThermoFisher, Waltham, MA) to a final volume of 90 μL. The Elongation Solution 1 was heated to 95 °C for 5 min and then 25 °C for 20 min. Elongation Solution 2 components: 50 μL of 10× PCR buffer (Denville Scientific, Charlotte, NC), 10 μL of 10 mM dNTP mix (New England Biolabs, Ipswich, MA), 4 μL of Choice Taq (Denville Scientific, Charlotte, NC), and diH2O to a final volume of 410 μL. The Elongation Solution 2 was heated to 95 °C for 5 min then cooled to room temperature for 20 min. Elongation Solutions 1 and 2 were combined and placed in a thermal cycler. Cycling conditions for elongation were as follows: 72 °C for 30 min; 25 °C for 10 min; and held at 4 °C. Step 2: To generate the round 0 RNA library, the round 0 dsDNA template was transcribed in vitro using the mutant Y639F T7 RNA polymerase. The in vitro transcription reaction conditions were as follows: a total reaction volume of 250 μL was composed of 250 pmol of the dsDNA template library, 25 μL of a 10× rNTP solution (10× rNTP solution: 10 mM 2′-OH purines (Roche, Indianapolis, IN; GTP and ATP) and 30 mM 2′-Fluoro pyrimidines (TriLink BioTechnologies, LLC, San Diego, CA; 2′-Fluoro-2′-dCTP, 2′-Fluoro-2′-dUTP)) with 50 μL of a 5× Transcription Buffer (5× Transcription Buffer: 20% w/v Peg-8000 (Sigma-Aldrich, St. Louis, MO), 200 mM Tris-HCL, pH

8.0 (Life Technologies, Carlsbad, CA), 60 mM MgCl2 (Sigma-Aldrich, St. Louis, MO), 5 mM spermidine HCl (Sigma-Aldrich, St. Louis, MO), 25 mM DTT (Sigma-Aldrich, St. Louis, MO), 2 μL of Inorganic pyrophosphatase (IPPI), ThermoFisher Scientific, Waltham, MA)) and diH2O to 250 μL. The in vitro transcription reaction was incubated overnight at 37 °C. Removal of the dsDNA template was performed by addition of 1 μL (10 units) DNase I (Roche, Indianapolis IN) for 30 min at 37 °C followed by chloroform extraction. Step 3: The in vitro transcribed round 0 RNA was purified using PAGE. The RNA was run on a denaturing PAGE gel (10% acrylamide (Bio-Rad, Hercules, CA); 8 M urea (RPI, Mount Prospect, IL)) for 30 min at 24 watts. The round 0 RNA band was detected by ultraviolet (UV) shadowing, excised and eluted from the gel with TE low-EDTA buffer (0.1 mM EDTA (Sigma-Aldrich, St. Louis, MO), 10 mM TE pH 7.5 (Sigma-Aldrich, St. Louis, MO)) for 1 h at 37 °C in a rotator (Labnet, Edison, NJ). The eluted round 0 RNA was run through a 0.2 μM Cellulose Acetate Centricon (Sigma-Aldrich, St. Louis, MO) to remove any residual gel fragments. A 10,000 Molecular Weight Cut-Off Amicon Ultra-4 Centrifugal Filter Unit (Millipore, Burlington, MA) was used to concentrate the RNA solution. RNA OD260 and 260/280 ratio were determined by NanoDrop 2000 (ThermoFisher, Waltham, MA) to assess quantity and purity of round 0 RNA library.

For each round of selection, we performed a negative selection step (to remove nonspecific binding aptamers) and positive selection step (to enrich for histone binding aptamers). The round 0 RNA library was first incubated with human IgG (negative selection) (Sigma-Aldrich, St. Louis, MO) in 1× binding buffer (BB) at 37 °C ((1× BB: 20 mM HEPES (Sigma-Aldrich, St. Louis, MO), 0.15 M NaCl (Sigma-Aldrich, St. Louis, MO), 2 mM CaCl2 (Sigma-Aldrich, St. Louis, MO)). Human IgG and human IgG binding aptamers were removed using a 0.2 μm nitrocellulose filter (GE, Chicago, IL). This was performed by incubating a 2 cm × 2 cm filter with the RNA solution for 20 min at 37 °C. Unbound RNA aptamers were transferred to a fresh tube containing (1000 pmol) of either human histone H3 or H4 (New England Biolabs, Ipswich, MA) and were incubated at 37 °C for 10 min. The histone-bound RNA aptamers were isolated by capturing the histones and RNAs on a nitrocellulose filter as described above. Histone-bound aptamers were eluted from the nitrocellulose filter by extraction with phenol/chloroform/isoamyl alcohol extraction. Specifically, the nitrocellulose disk containing the histones and histone-bound aptamers was submerged in phenol/chloroform/isoamyl alcohol (Fisher Scientific, Hampton, NH) and vortexed for 1 min. The solution was separated into two phases by adding an equal volume of diH2O, vortexed for 1 min and centrifuging at $17,000 \times g$ for 10 min. The aqueous layer (top layer containing the RNA aptamers) was collected and placed into an RNAse and DNAse-free 1.5 mL microcentrifuge tube (USA Scientific, Ocala, FL). An equal volume of chloroform (Fisher Scientific, Hampton, NH) was added to the aqueous layer to remove any trace of phenol, the solution was vortexed for 1 min, and centrifuged at $17,000 \times g$ for 10 min. The aqueous layer (top layer containing the RNA aptamers) was collected and placed into a clean 1.5 mL microcentrifuge tube. The RNA aptamers were then ethanol precipitated by adding 5 μL of linear acrylamide (ThermoFisher, Waltham, MA), 1/10 volume of 10 M ammonium acetate (Fisher Scientific, Hampton, NH) and 2.5 volume of 100% ethanol (Pharmco-Aaper, Shelbyville, KY) followed by a 2 s vortex and centrifugation at $17,000 \times g$ for 15 min. The solution was incubated at −20 °C overnight, and then centrifuged for 15 min at $17,000 \times g$. The supernatant was removed, and the pellet washed with 1 mL of 95% ethanol. The solution was centrifuged at 4 °C for 5 min at $17,000 \times g$, and the supernatant removed and the RNA pellet air dried at room temperature. The RNA pellet was then resuspended in 25 μL of diH2O. The RNA was then reverse-transcribed by combining 10 μL of 5× FS buffer (ThermoFisher Scientific, Waltham, MA), 1 μL 0.1 M DTT (Sigma-Aldrich, St. Louis, MO), 1 μL 100 μM Sel2 3′primer (5′-TCGGGCGAGTCGTCTG-3′ (IDT, Coralville, IA)), 31 μL diH2O and 5 μL of recovered RNA in a polymerase chain reaction (PCR) tube (USA Scientific, Ocala, FL). Reaction conditions are as follows: 65 °C for 5 min and 72 °C for 5 min (annealing step) and room temperature for 10 min (cooling step). Then, 1 μL of 10 mM dNTP mix and 1 μL of Superscript III Reverse Transcriptase (ThermoFisher Scientific, Waltham, MA) was added to the mix followed by an incubation of 60 min at 55 °C, then 72 °C for 15 min and cooled to 4 °C. Next, 25 μL of the transcribed material was added to 50 μL of 10× Taq Polymerase Buffer (Denville Scientific, Charlotte, NC), 20 μL of 10 mM dNTP mix, 5 μL of 100 μM Sel2 5′ primer, 5 μL of 100 μM Sel2 3′-primer, 25 units of Taq polymerase (Denville Scientific, Charlotte, NC), and brought up to 500 μL with diH2O. The reaction was separated into five PCR tubes and ran at 95 °C for 2 min, 22–25 cycles of 95 °C for 30 s, 55 °C for 30 s and 72 °C for 5 s, followed by 72 °C for 5 min, and then held at 4 °C. The amplified DNA (round 1 DNA) was utilized as the template for the subsequent round of selection as described above to generate the round 1 RNA pool. Selection conditions for each round of selection are provided in Supplemental Table 1.

**Double-filter binding assay**. Double-filter nitrocellulose binding assay was preformed to determine the binding affinity ($K_D$) of the aptamers for their target. Briefly, the transcribed RNA from RD0 and RD8 of selection were dephosphorylated using Bacterial Alkaline Phosphatase (ThermoFisher, Waltham, MA). The dephosphorylating conditions were as follows: 100 pmol of RD0 or RD8 RNA was combined with 5 μL of 1 M Tris-HCL, pH 8.0, 1 μL BAP, and supplemented with diH2O to reach 100 μL. The solution was then incubated at 55 °C for 60 min,

followed by adding 10 μL of 10× Dephosphorylation Stop Mixture ((20 mM Tris-HCl, pH 8.0, 40 mM EDTA, 200 mM NaCl (RPI, Mount Prospect, IL), 1% SDS (RPI, Mount Prospect, IL) (w/v)). Then, 300 μL of low-EDTA buffer was added to the reaction and purified using the phenol/chloroform/isoamyl alcohol extraction described above.

The dephosphorylated RNA and synthetic aptamers (KU7 and KU9) were radiolabeled with gamma-$^{32}$P-ATP using T4 polynucleotide kinase (PNK) as follows. A total reaction volume of 20 μL was made up of 20 pmol of aptamer RNA, 2 μL PNK (New England Biolabs, Ipswich, MA), 2 μL PNK reaction buffer (New England Biolabs, Ipswich, MA), 2 μL gamma-$^{32}$P-ATP (PerkinElmer, Waltham, MA) and diH$_2$O. The mixture was then incubated at 37 °C for 30 min and 65 °C for 20 min to heat inactivate the PNK. Then, 20 μL of 1× BB was added to the reaction followed by a centrifugation step through a G25 purification column (GE Healthcare, Little Chalfont, UK) according to the manufacturer's instructions. Labeling efficiency was determined by a scintillation counter. All radiolabeled RNAs were diluted in 1× BB to 2000 cpm mL$^{-1}$. Then, 5 μL of 2000 cpm mL$^{-1}$ radiolabeled RNA were incubated at 37 °C for 5 min with 15 μL of either human histone H3.2 or H4 (positive selection targets) or human albumin (Sigma-Aldrich, St. Louis, MO) or serum (Sigma-Aldrich, St. Louis, MO) (negative control proteins) serially diluted in 1× BB. The binding reactions were loaded onto a dot blot apparatus (composed of nitrocellulose membrane on the top, nylon membrane (Sigma-Aldrich, St. Louis, MO) in the middle and Whatman paper (Sigma-Aldrich, St. Louis, MO) on the bottom). Treatment of the nitrocellulose membrane was as follows: pretreated with 0.5 M KOH (Sigma-Aldrich, St. Louis, MO) for 20 min, quick wash with diH$_2$O and equilibrated in 0.1 M Tris-HCl 7.5 for 45 min, washed with diH$_2$O and transferred to 1× BB for 20 min before l (Sigma-Aldrich, St. Louis, MO). The nylon was also incubated in 1× BB for 20 min before being loaded on the manifold. Before loading the RNA/protein samples, the wells were washed with 100 μL of 1× BB. The amount of RNA bound (nitrocellulose) versus unbound (nylon) was determined by densitometry of imaged membrane on a Fuji Phosphor imager.

**Illumina high-throughput sequencing and sample preparation.** The RNA pools from all rounds of selection were reverse-transcribed using Superscript III Reverse Transcriptase and the Sel2 3′ primer. Briefly, the RNA from each selection round was heated for 60 min at 55 °C followed by 72 °C for 15 min. The DNA product was then PCR-amplified using Choice Taq DNA Polymerase with barcoded Illumina primers (Supplementary Table 4). The DNA product with the Illumina barcode was heated for 2 min at 95 °C, followed by 10 cycles of 95 °C for 30 s, 55 °C for 30 s and 72 °C for 30 s with a final extension step at 72 °C for 5 min. The PCR product was then run on a 2% agarose gel and the appropriate bands (approximately 151 base pairs) were excised. The gel fragment was purified using the Qiagen Gel Extraction kit (Hilden, Germany) and quantified using NanoDrop 2000. Qualitative analysis of the samples was performed on an Agilent Model 2100 Bioanalyzer (Agilent, Santa Clara, CA). The samples were combined at equal molar amounts and submitted for Illumina sequencing (University of Iowa Genomic core, Iowa City, IA; Illumina Genome Analyzer II)[64].

The Illumina base calls (sequence reads) were pre-processed and filtered to identify the Sel2N20 aptamer library variable region sequence. For each round of selection, all unique variable region sequences were enumerated with the repeat sequences being counted. The sequenced DNA code was converted to RNA and the 5′ and 3′ constant regions were added.

**Surface plasmon resonance.** pH scouting was performed with all proteins. Coating of histone H4 and BSA to the COOHV biosensor was performed at pH 4.7. Coating of calf thymus histone to the COOHV biosensor was performed at pH 6.0. To prevent nonspecific binding to the vial and sensor surface, a small concentration (~0.01%) of Tween-20 was added to these solutions. The proteins were immobilized following a standard amine coupling procedure where a mixture of EDC (1-ethyl-3-(3-dimethylaminopropyl) carbodiimide) and NHS (N-hydroxysuccinimide) was injected to activate the surface carboxyl groups and protein was injected to react with and be coupled to the sensor surface. Then, 1 M Tris pH 8 was injected last to deactivate remaining active carboxyl groups.

A twofold dilution series (ranging from 10 to 100 nM) was prepared for each aptamer in buffer. Each sample was injected over all three flow channels for 4 min at a flow rate of 30 μL min$^{-1}$ and dissociation was monitored for 10 min. The sensor surface was regenerated after each analyte cycle by injecting 1 M NaCl for 1 min. Data were processed by subtracting the reference channel signal and buffer blank responses (double referencing method). A two-compartment kinetic (mass transport limited) model was fit to the data to determine rate constants of association ($k_a$) and dissociation ($k_d$). A local $R_{max}$ was provided in the fit due to an apparent concentration-dependent surface activity.

**Serum stability measurements.** Human blood was collected from healthy volunteers after obtaining written informed consent with protocols conducted in accordance to global ethical standards and approved by the University of Iowa Institutional Review Board. Chemically synthesized aptamers KU7 and KU9 were incubated at 37 °C under 5% CO$_2$ at a concentration of 5 μmol L$^{-1}$ in 50% human serum. Incubation times ranged from 0 min to 28 days. At each time point, 10 pmol

of aptamer (2 μL) was removed and added to 8 μL of 1× Tris base, Boric Acid, EDTA (TBE) (Fisher Scientific, Hampton, NH) and 2 μl of 2× Urea RNA loading dye (0.01 g Xylene Cyanol, 0.01 g bromophenol blue (Sigma-Aldrich, St. Louis, MO), 500 μl of 10× TBE and 10 mL of formamide (Amresco, Solon, OH)). The samples were then heat-inactivated at 95 °C for 5 min and stored at −20 °C. All samples (10 μL per lane = 8 pmol aptamer RNA) were loaded into a denaturing PAGE gel (8 M urea 10% acrylamide) and separated by electrophoresis at 10 watts for 30 min. The gel was stained with SYBR Gold (Invitrogen, Carlsbad, CA) at a 1:10,000 dilution in phosphate-buffered saline (PBS) for 30 min and visualized by UV light using a Chemidoc (Bio-Rad, Hercules, CA). Briefly, the quantification area of the aptamers at each time point was set to the height and size of the band at the 0 min time point. The band intensity of each time point was then calculated and normalized to 0 min, which was set at 100%.

**Platelet isolation.** Healthy donor blood (20 mL) was collected in 3.2% sodium citrate (9:1, v/v) vacutainer tubes (BD Scientific, San Jose, CA). The blood was pooled into 15 mL conical tubes (CellTreat, Pepperell, MA) and centrifuged at $100 \times g$ for 15 min (without brakes) at room temperature. Platelet-rich plasma (PRP) was collected into fresh tube and prostaglandin E1 (PGE1, final conc.1 μM) was added, and mixed gently by inverting tubes. PGE1-PRP was then centrifuged at $800 \times g$ for 10 min to pellet the platelets. After centrifugation, platelet pellets were washed with modified Tyrode's buffer (134 mmol L$^{-1}$ NaCl, 2.9 mmol L$^{-1}$ KCl, 2.9 mmol L$^{-1}$ CaCl$_2$, 0.34 mmol L$^{-1}$ Na$_2$HPO$_4$, 12 mmol L$^{-1}$ NaHCO$_3$, 20 mmol L$^{-1}$ HEPES, 1.0 mmol L$^{-1}$ MgCl$_2$, 5.0 mmol L$^{-1}$ glucose, pH 7.35) containing 1 μM PGE1. After washing, the platelet pellets were resuspended in modified Tyrode's buffer and counted on ADVIA 120 Hematology System (Siemens Healthineers, Malvern, PA).

**Platelet aggregation assay.** Washed platelets were prepared as above and resuspended in Tyrode buffer to a final concentration of $2.5 \times 10^8$ platelets per mL. For the platelet aggregation studies, 400 μL of washed platelets were stirred at 1200 rpm at 37 °C along with collagen (1 μg mL$^{-1}$), histone H4 (10 μg mL$^{-1}$), selected RNA rounds and/or KU7 or KU9 in a cuvette of an aggregometer (Chrono-log Model 560-VS) and light transmittance was recorded. Aggregation was measured as percent change in light transmission, where 100% refers to transmittance through blank solution.

**Neutrophil isolation.** Healthy donor blood (50 mL) was collected in 3.2% sodium citrate (9:1, v/v) vacutainer tubes (BD Scientific, San Jose, CA). The blood was mixed with equal volume of room temperature dextran/saline solution in 50 mL conical tubes (CellTreat, Pepperell, MA). The solution was then mixed by inverting 10 times and incubating upright for 20 min at room temperature. The upper layer (leukocyte-rich plasma) was carefully collected and transferred into a new 50 mL conical tube. The leukocyte-rich plasma solution was then centrifuged at 5 °C for 10 min at $250 \times g$. The supernatant was discarded and the cells were immediately resuspended in 30 mL of 0.9% saline. A 10 mL ficoll-hypaque solution (Sigma-Aldrich, catalog number: 10771) was added beneath the cell suspension and then centrifuged at 20 °C for 30–40 min at $400 \times g$ with no breaks. The top layer was aspirated leaving the neutrophil and red blood cell (RBC) pellet behind. The cell pellet was resuspended in 20 mL of ice-cold water for exactly 30 s followed by 20 mL of cold 1.8% NaCl solution to restore isotonicity (0.9%). The neutrophil-containing solution was then centrifuge at 5 °C for 6 min at $250 \times g$. If the pellet was white, we resuspended in 5–10 mL of ice-cold PBS and counted the cells. If the pellet was red, we repeated the ice-cold water treatment as detailed above to remove RBC.

**NETosis induction.** Neutrophils were plated on 150 cm × 150 cm × 20 mm tissue culture dishes. After 30 min, when cells had adhered to the bottom of the dishes, they were stimulated for 4 h with 500 nM PMA. Conditioned medium was gently aspirated and the neutrophil/NET monolayer was collected with ice-cold PBS-Ca$^{++}$/Mg$^{++}$ and centrifuged for 10 min at $450 \times g$ at 4 °C to sediment cells. The cell-free but NET-rich supernatant was centrifuged for 10 min at $18,000 \times g$ at 4 °C and the pellet containing NETs was suspended in Opti-MEM and DNA quantified by measuring absorbance at 260 nm using a NanoDrop spectrophotometer.

**Imaging of NETs.** To visualize NET formation, neutrophils were seeded on 12 mm coverslips inserted into 12-well plates (300,000 cells per well)[65]. After cells had adhered, they were stimulated with PMA as described above and aptamer binding to NET-associated chromatin was visualized by immunofluorescence[66]. Briefly, after fixing with 4% paraformaldehyde and permeabilizing with 0.5% Triton X-100, coverslips were blocked with 5% normal goat serum overnight at 4 °C. They were then sequentially incubated for 2.5 h each with anti-histone H3 (Cayman #13784, 1:50) antibody and 2 μM KU7–647 aptamer diluted in blocking buffer. After washing with PBS, cells/NETs were incubated with goat anti-rabbit IgG-Alexa Fluor 488 (1:360) for 1 h and counterstained with 1.33 μg mL$^{-1}$ DAPI. All incubations were performed in a humidified chamber at 37 °C and were followed by three 5 min washes in PBS. Cells were mounted onto glass slides using Vectashield with DAPI. Images were captured using Leica LMD7000 confocal microscope under 40× oil per 2.8× zoom magnification and processed using Fiji Imaging software.

**Cytotoxicity assays**. EA.Hy926 cells were seeded in a 96-well flat bottom plate (CellTreat, Pepperell, MA) at a density of 8,000 cells per well in 100 μL of medium. After 24 h the medium was removed and replaced with 50 μL of Opti-MEM (GIBCO, Gaithersburg, MD) containing either calf thymus histones (Sigma-Aldrich, St. Louis, MO) and/or aptamer. After 24 h the medium was removed and MTS reagent (Abcam, Cambridge, MA) was added for 1 h according to manufacturer's protocol and quantified using a Thermo Max Microplate Reader (Molecular Devices, Sunnyvale, CA) at 490 nm.

EA.Hy926 cells were seeded in a 96-well flat bottom plate (CellTreat, Pepperell, MA) at a density of 8000 cells per well in 100 μL of medium. After 24 h, the medium was removed and replaced with 50 μL of Opti-MEM (GIBCO, Gaithersburg, MD) containing 8 μg of NET material (determined by DNA concentration) and/or 8 μg (10.66 μM) of aptamer. After 24 h, the medium was removed and MTS reagent (Abcam, Cambridge, MA) was added for 1 h according to the manufacturer's protocol and quantified using a Thermo Max Microplate Reader (Molecular Devices, Sunnyvale, CA) at 490 nm.

EA.Hy926 cells were seeded in a 96-well flat bottom plate (CellTreat, Pepperell, MA) at a density of 16,000 cells per well in 100 μL of medium. After 24 h, the medium was removed and replaced with 40 μL of Opti-MEM (GIBCO, Gaithersburg, MD) containing 200 μg mL$^{-1}$ calf thymus histones (Sigma-Aldrich, St. Louis, MO). Quadruple wells received 10 μL of either vehicle (negative control), heparin (positive control, 1:1), KU7 aptamer (1:2) or KU9 aptamer (1:2) at time points 0, 5, 10, 15, 30, 45, 60, 90, 120 and 180 min. After 3 h, all wells were washed with media, followed by incubation in MTS reagent (Abcam, Cambridge, MA) for 1 h according to the manufacturer's protocol and quantified using a Thermo Max Microplate Reader (Molecular Devices, Sunnyvale, CA) at 490 nm.

**Intracellular calcium influx measurements**. Intracellular calcium concentration ($[Ca^{2+}]i$) was measured in EA.hy926 cells using ratiometric calcium indicator Fura 2-AM (Life Technologies, Carlsbad, CA). Cells grown on 35 mm glass-bottom plates (MatTek, Ashland, MA) were loaded with 1 μM Fura 2-AM in calcium buffer (140 mM NaCl, 2.8 mM KCl, 2 mM CaCl$_2$, 2 mM MgCl$_2$, 10 mM glucose and 10 mM HEPES) for 15 min, washed and incubated in calcium buffer for another 15 min. Fura 2-AM-loaded cells were illuminated in calcium buffer by alternating 340/380-nm light delivered every 300 ms by a DG-4 argon exciter (Sutter Instruments, Novato, CA) under the control of MetaFluor software, and fluorescence images were captured at an emission of 510 nm with a Photometrics Cool SNAP HQ charge coupled device camera (Roper Scientific, Tucson, AZ) based on a Nikon TE2000 fluorescent microscope. Calf thymus histones (50 μg mL$^{-1}$) without or with 3 min of pre-incubation with RNA aptamer KU7 at various concentrations were added to loaded cells and calcium images were recorded for 10 min for each treatment condition. $[Ca^{2+}]i$ was reported as the ratio of Fura 2 fluorescence emission at 340 and 380 nm (F340/F380) normalized to baseline. All procedures were performed at room temperature.

**TLR activation studies**. TLR activation as measured by IL-6 production. EA. Hy926 cells were seeded in a 96-well flat bottom plate at a density of 15,000 cells per well in 100 μL of medium. After 24 h, the medium was removed and replaced with 100 μl of Opti-MEM containing either calf thymus histones and/or aptamers. After another 24 h, the supernatant was collected, processed and quantified according to the manufacturer's protocol used in the human IL-6 Quantikine® ELISA kit (R&D Systems, Minneapolis, MN).

**Animals**. The 8- to 12-week-old BALB/cJ mice (Jackson Laboratory, Bar Harbor, ME) were used for this study. All procedures conformed to standards established in the Guide for Care and Use of Laboratory Animals (National Academy Press, Washington, D.C. 2011). The Institutional Animal Care and Use Committees of the University of Iowa, accredited by AAALAC (Association for Assessment and Accreditation of Laboratory Animal Care International), reviewed and approved all protocols. All efforts were made to minimize the number of animals used and to avoid experiencing pain or distress.

**Mouse model of multiple organ dysfunction**. Mice received a tail vein injection of calf thymus histones at 50 mg kg$^{-1}$ and/or aptamer at 25 or 12.5 mg kg$^{-1}$ or vehicle (100 μL per mouse). The mice were monitored for distress and were killed 6 h post injection with isoflurane overdose if they survived histone toxicity.

Mice received retro-orbital injection of calf thymus histones at 62.5 mg kg$^{-1}$ or vehicle (100 μL per mouse). After 30 min, mice received a follow-up retro-orbital injection (opposite eye) of either vehicle or aptamer at 31.25 mg kg$^{-1}$ (100 μL per mouse). The mice were monitored for distress and were killed 6 h post injection with isoflurane overdose if they survived histone toxicity.

Mouse weights were recorded prior to tail vein injections using a digital laboratory balance (Denver Instrument MXX-601, Bohemia, NY). Organs were harvested and weighed after killing and normalized to the vehicle-treatment weight of each animal.

Histology was performed at the Comparative Pathology and Histology Research Laboratories (University of Iowa; http://www.medicine.uiowa.edu/pathology/research/dcp/). Tissues (lung, liver and spleen) were excised and fixed in 4% paraformaldehyde (PFA) (Affymetrix, Santa Clara, CA) for at least 1 week at 4 °C.

All fixed tissues were processed in a series of alcohol and xylene baths, paraffin embedded and 7 μm sections were stained with H&E. A veterinary pathologist (David K. Meyerholz) examined all tissue sections. Dr. Meyerholz was blinded to the sample IDs. High-resolution digital images were acquired with a DP71 camera (Olympus) mounted on a BX51 microscope (Olympus) with MicroSuite Pathology Edition Software (Olympus).

Organs (lung, liver and spleen) were dissected out immediately after death or killing. A block of tissue (5 mm × 5 mm × 5 mm) from each organ was fixed in 4% PFA for 5 h and then cryo-protected overnight in 30% sucrose in PBS at 4 °C. Frozen 20 μm coronal sections were cut with a cryostat and mounted on Colorfrost Plus microscope slides (Fisher Scientific, Hampton, NH). Procedures like those described in previous publications were used for different combinations of multiple-label immunofluorescent staining for histone H3, myeloperoxidase (MPO, a marker for activated neutrophils), CD41 (a marker for platelets) and IL-6, and fluorescent staining for cytoskeleton (phalloidin-Alexa Fluor 568) and nucleus (TO-PRO®−3). Briefly, sections were incubated in 10% donkey serum with a mixture of primary antibodies that were made in different species. The antibody for histone H3 was purchased from Cell Signaling (Danvers, MA, dilution 1:200), MPO from Abcam (Cambridge, MA, dilution 1: 200), CD41 from BD Biosciences (San Jose, CA, dilution 1:400) and IL-6 from Novus Biologicals (Littleton, CO, dilution 1:100). After washing with PBS, sections were incubated in a mixture of fluorophore-conjugated secondary antibodies (Alexa Fluor 488 conjugated donkey anti-rabbit IgG and/or Alexa Fluro 594 conjugated donkey anti-rat IgG, both from Jackson ImmunoResearch, Westgrove, PA; dilution 1:200) against species from which the primary antibodies were made. Stained sections were covered using coverslips and Prolong Diamond Anti-Fade Reagents (Invitrogen, Carlsbad, CA) after the final washes with 1× PBS. Negative controls consisted of tissue processed in the absence of primary antibodies.

Fluorescent immuno-stained slides were examined with a Zeiss LSM 710 confocal laser scanning microscope. Sections were scanned sequentially in different channels to separate labels. Images from different channels were each assigned a pseudo-color and then were superimposed. Confocal images were obtained and processed with software provided with the Zeiss LSM 710.

Mouse organs (liver, lungs and spleen) were excised and immediately flash frozen. The organs were ground using a mortar and pestle (RPI, Mount Prospect, IL) over dry ice, followed by total cellular RNA extraction using the RNeasy Mini Kit (Qiagen, Valencia, CA). The purity and concentration of isolated RNA was measured using a NanoDrop 2000c spectrophotometer (ThermoFisher Scientific, Waltham, MA). The RNA was then reverse-transcribed into complementary DNA using SuperScript III Reverse Transcriptase (ThermoFisher Scientific, Waltham, MA). Real-time PCR was used to quantify the mRNA expression levels of the immune-responsive gene *IL-6*. Briefly, 100 ng of cDNA was used per 20 μL reaction of Power SYBR Green (Applied Biosystems, Foster City, CA). All PCR cycle programs were as follows: 95 °C for 10 min (95 °C for 15 s, 60 °C for 50 s) × 40 cycles, followed by a melting curve (95 °C for 15 s, 60 °C for 1 min, 95 °C for 1 s). Each reaction was performed in triplicate, and relative gene expression data were calculated as fold change to β-actin (control) expression data. The mouse-specific primers used in this study were: *β-Actin* (forward: CGG TTC CGA TGC CCT GAG GCT CTT; reverse: CGT CAC ACT TCA TGA TGG AAT TGA), *IL-6* (forward: GAG GAT ACC ACT CCC AAC AGA CC; reverse: AAG TGC ATC ATC GTT GTT CAT ACA).

**Statistical analysis**. Results are expressed as mean± SEM. Statistical comparisons were performed by a two-tailed *t*-test, or one-way or two-way analysis of variance (ANOVA) with correction for multiple comparisons using GraphPad Prism v.7.04. A *p* value of <0.05 was considered significant.

**Reporting summary**. Further information on experimental design is available in the Nature Research Reporting Summary linked to this article.

## Data availability
The datasets generated during and/or analyzed during the current study are available from the corresponding author on request.

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

## Acknowledgements

The authors would like to acknowledge the use of the University of Iowa Pathology Core and the University of Iowa Central Microscopy Research Facility, resources supported by the Vice President for Research & Economic Development, the Holden Comprehensive Cancer Center and the Carver College of Medicine. We would like to thank the labs of Drs. Adam Dupuy, George Wiener and the Iowa City VA Microscopy Core for the use of equipment (Bio-Rad Imager, plate reader and confocal microscope). K.T.U. was supported by the American Heart Association (AHA) pre-doctoral fellowship (17PRE33410335). This work was supported by grants to P.H.G. and F.J.M. from the Department of Defense Congressionally Directed Medical Research Programs (PR150627 and PR150627P1), University of Iowa Carver College of Medicine (CCOM) (Carver Collaborative Pilot Grant Award 2015) and University of Iowa Award from The Office of the Vice President for Research and Economic Development (OVPRED 2015). F.J.M. is also supported by the Office of Research and Development, Department of Veterans Affairs (2I01BX001729). This study was also supported in part by the National Institutes of Health grants AG049784 to S.D. and HL121105 to J.K.-T.

## Author contributions

Conceptualization: F.J.M. and P.H.G.; methodology: F.J.M., P.H.G., S.D., M.E.W., J.A.S. and L.-H.L.; data acquisition, curation and analysis: K.T.U., G.N.B., L.-H.L., J.P.D., S.S., W.H.T., Y.C., V.K.S., B.L., J.A.S., S.M., W.R.G., J.K.-T., F.J.M. and P.H.G.; resources: F.J.M. and P.H.G.; writing (original draft): K.T.U., G.N.B. and L.-H.L.; writing (review and editing): J.K.-T., F.J.M., P.H.G., S.S. and L.-H.L.; supervision: F.J.M. and P.H.G.

## Additional information

**Competing interests:** The authors declare no competing interests.

