## [Peer Review File · Nature Communications]

Reviewers' Comments:

Reviewer #1:

Remarks to the Author:

The authors have used the SELEX system to produce anionic aptamers which they have shown to "neutralize" calf thymus histones (CTH), based on in vitro studies. These are quite novel findings. They have also shown that the aptamers are strongly protective against the lethal effects of infused CTH (containing H1, H2A, H2B, H3 and H4 histones). The aptamer in Figure 5C protected against the lethality in vivo after CTH infusion and reduced in increased $[Ca^{2+}]_i$ in EA.hy926 cells (human endothelia cell line) exposed to histones in vitro. The authors need to respond to the following concerns:

1) Why did the authors focus on effects of aptamers on H3 and H4 (Figure 1B, C) exclusive of the other histones? While it is not known precisely in mouse sepsis (after cecal ligation and puncture, CLP) which histones are present in plasma, and the histone neutralizing antibody (from Roche) is protective in polymicrobial mice. The current report restricts studies H3 and H4 raising the issue as to whether the aptamers (especially KU9) would be highly protective in the CLP model, and ultimately in human sepsis and to what extent the aptamers will also block H1, H2A and H2B histones. Such information is vital if the ultimate application of the aptamers is to treat humans with sepsis. To emphasize this point, a recent publication emphasizes that, in human sepsis, H2B and H3 are present (in ng/ml quantities in plasma), although other histones were not assessed (J. Garcia-Gimenez et al, Scientific Reports 7, 10643 (2017)). Furthermore, the authors could easily do these experiments. To determine if the aptamers bind to and block biological effects of H1, H2A, H2B.

2) It would be important to determine if the aptamers neutralize both "natural" histones and histones derived from neutrophils induced in vitro to form Nets in the presence or absence of PAD4, which converts arginine to citrulline, which occurs naturally during sepsis. The prediction would be that aptamers would work in both situations, but there are no available data. There are several PAD4 inhibitors that are commercially available and could be used for such studies. There data would be important for an understanding of what the aptamers are doing in vitro and in vivo.

Reviewer #2:

Remarks to the Author:

The authors describe efforts towards the development of novel aptamers that have high affinity for extracellular histones, which is implicated in the organ dysfunction syndrome, MODS. The first few pages of the manuscript are devoted to the pathology of MODS and limitations of current therapies. The authors make many points in this study. For example, they use 2'-fluoro modified RNA that effectively prevented RNA degradation induced by nuclease in human serum. Also, during the course of the RNA selection, the authors perform both positive and negative screenings in an attempt to impart specificity to binding to histone H3 versus H4 and no other proteins. The RNA aptamers inhibit histone H4-induced platelet aggregation in in vitro experiment, whereas they inhibit IL-6 production and increase of calcium concentration in vivo. The authors test intravenous injection of histone with/without the aptamer into mice where the aptamer increases mice survival, prevents vascular congestion, and improves survival of histone-injected mice.

My major concern is that there is no evidence for specificity of the aptamers. Round 0 and later-round aptamers work almost equally in binding histones, bind the non-cognate histone tightly, and promote platelet aggregation nearly equally. Moreover, binding is not at equilibrium, which may or may not explain the effects. Also, the primer binding sites are involved in aptamer formation. While some physiological effects might be interesting, it seems the same results could be obtained just by using 2'-fluorinated random RNA.

Referees' concerns:

Major Comments

1. The authors describe “a leftward shift in the binding curve of the R8 RNA pools compared to the R0 RNA pools for both selections is indicative of enrichment of high-affinity binding RNA sequences” at the first paragraph on Result section. The random RNA pools at the round 0, however, seem like they already have strong affinity for both histone H3 and H4. Is this because histone proteins have non-specific affinity for nucleic acids?

The authors should plot their Figure 1B data on a logarithmic x-axis to spread out the data near the origin to help the reader judge any differences. Moreover, logarithmic concentrations are linear in free energy and allow the reader to look for sigmoidal behavior and judge the Hill coefficient. In such a plot the zero protein concentration is omitted and can be added with an axis break if desired.

2. The above concern is reinforced the aggregometer readings in Supp Fig 3 where R0 aptamers have nearly identical efficacy in preventing aggregation with time as R3/R5 aptamers.

3. p7. Specificity of the aptamers is further called into question also by the fact that KU7 and KU9 bind to H3 AND H4 even though they were selected for binding to H3 only.

4. On a related note, the size of the library, which should be mentioned in the main text, is only 20 nt. This is quite small for such a large protein.

5. p7. This reader does not understand the sentence, “The top three RNA sequences from each selection (H3 selection: KU4 – KU6; H4 selection: KU7 – KU9) were selected based on sequence enrichment between rounds 3 to 5 (positive selection) and rounds 6 to 8 (negative selection).” Why is rounds “3 to 5” followed by “(positive selection)” and “rounds 6 to 8” followed by “(negative selection)”. Also, is the first part of this sentence meant for H3 selection and the second part for H4 selection? Also, why the abbreviation “KU”?

6. The time for equilibration of protein and aptamer was only 5 minutes. At low concentrations of aptamer this is not enough time for binding. e.g. Aptamer was reported in the SPR in Fig 2 to have an on-rate constant of $1.6 \times 10^5 \text{ M}^{-1} \text{ s}^{-1}$. At 1 nM of aptamer, which is in Figure 1D and would be needed for K_d of 1 nM as in Supp Table 2, would need on the order of 6000 sec $[=1/(1.6 \times 10^5 \text{ M}^{-1} \text{ s}^{-1} \times 1 \times 10^{-9} \text{ M})]$ or 100 min (~2h) to bind to completion. Since this length of time wasn't tested, there is no way to know if the system is at equilibrium. The authors must assure equilibrium by showing they get the same answer with longer times of equilibration. Indeed, this may be the root cause of why binding curves H3 R0 and H3 R8 are essentially superimposable in Fig 1B.

Along these lines, In Fig. 1B, Fig. 2 and Supplementary Fig. 2A, the authors should consider appropriate range of higher histone-concentration (i.e. concentrations chosen are too high). In Fig. 1B and Supplementary Fig. 2A, the fraction bound of aptamers (in particular, H3R8, H4R0, KU7 for H3.2 and H4, and KU9 for H4) reaches at plateau at second lowest concentration of histones. In Fig. 2, estimated K_D values are out of the range that they tested, which means the test condition is less quantitative. Appropriate range is in a range of lower histone concentration.

7. p7. The authors state that the reasoned that two of the aptamers would be stable but don't explain their reasoning.

8. Supp Fig 1. What is the meaning of the different colors? Why are there two structures provided for each sequence? Why are the flanking regions involved in the aptamer? Does the fact that they are interacting with the random region and perhaps more importantly with themselves at the base of every structure mean that the random region wasn't long enough? Or perhaps there is no specificity and any random 2'F RNA would have the same effect.

9. Despite the authors evaluate RNA stability in human serum in Fig. 2B, they don't show the corresponding gel image in KU5 in Supplementary Fig. 2B. Also, what is the reader looking at in Supp Fig 2B? Is this the top of a gel? If so, the degradation down on the gel should be shown. Or is this a slot blot? If so, the retention of various RNA lengths should be provided because there could be complete degradation and still lots of signal.

10. p14. The argumentation for 2'F does not make good chemical sense. It makes sense that 2'F increases duplex stability, by locking in the proper sugar pucker of 2'-exo/3'-endo for an A-form duplex, as the authors state. But the aptamers in the present study bind proteins not other RNAs, and there is no reason a priori to think 2'-exo sugar pucker are needed for this. More likely is that resistance to nuclease-mediated and chemical degradation, which rely on the 2'OH, is offered by the 2'F.

Other Comments

11. p4. It is not wholly accurate to say that aptamers do not aggregate. Aggregation of RNA by base pairing or quadruplex formation is well known.

12. p5. High affinity (nM Kd) does not always lead to "an excellent reagent". Specificity plays an equally important role as affinity in the process.

13. p11. The text mentions that mice were treated with KU9 at 30 min and references Fig 5C. But Fig 5C shows KU9 administered at 180 and 260 min only.

14. Fig 5A suggests that Vehicle blocks cell survival which doesn't make sense. Are the symbols on Vehicle and Heparin switched?

15. Page 10. "Death occurred between one to three hours-----of aptamer (Fig. 4A and 5B)". This should be Fig. 4A and 5C. Following sentence "Injection of histones-----by the aptamer treatment (Fig. 4B and 5C)". This should be Fig. 4B and 5D.

16. p12. Remove comma between "which" and "would".

17. In Fig. 3C, CTH concentrations should be represented by molar concentration. it is helpful to figure out molecular ratios between aptamer and CTH.

18. The authors should provide unit in Supplementary Table 2.

19. Page 10. "Death occurred between one to three hours-----of aptamer (Fig. 4A and 5B)". This should be Fig. 4A and 5C. Following sentence "Injection of histones-----by the aptamer treatment (Fig. 4B and 5C)". This should be Fig. 4B and 5D.

20. p12. Remove comma between "which" and "would".

21.

Rebuttal

We thank the reviewers for their detailed and helpful comments on our manuscript. We have addressed each of the comments and revised the manuscript accordingly. Changes in the manuscript text are highlighted using track changes. Excerpts from the reviewers' critiques are indicated below in *italics*.

Reviewer #1 (remarks to the author)

The authors have used the SELEX system to produce anionic aptamers which they have shown to "neutralize" calf thymus histones (CTH), based on in vitro studies. These are quite novel findings. They have also shown that the aptamers are strongly protective against the lethal effects of infused CTH (containing H1, H2A, H2B, H3 and H4 histones). The aptamer in Figure 5C protected against the lethality in vivo after CTH infusion and reduced in increased [Ca²⁺] in EA. hy926 cells (human endothelia cell line) exposed to histones in vitro. The authors need to respond to the following concerns:

Major comments

- 1) Why did the authors focus on effects of aptamers on H3 and H4 (Figure 1B, C) exclusive of the other histones? While it is not known precisely in mouse sepsis (after cecal ligation and puncture, CLP) which histones are present in plasma, and the histone neutralizing antibody (from Roche) is protective in polymicrobial mice. The current report restricts studies H3 and H4 raising the issue as to whether the aptamers (especially KU9) would be highly protective in the CLP model, and ultimately in human sepsis and to what extent the aptamers will also block H1, H2A and H2B histones. Such information is vital if the ultimate application of the aptamers is to treat humans with sepsis. To emphasize this point, a recent publication emphasizes that, in human sepsis, H2B and H3 are present (in ng/ml quantities in plasma), although other histones were not assessed (J. Garcia-Gimenez et al, Scientific Reports 7, 10643 (2017)). Furthermore, the authors could easily do these experiments. To determine if the aptamers bind to and block biological effects of H1, H2A, H2B.*

Histones H3 and H4 were chosen because these are the major histone isoforms implicated in MODS (including MODS from sepsis). However, as noted by the reviewer, recent findings have also implicated other histone subunits (e.g. H2B) in sepsis-induced MODS. We now include new binding studies for all histone subunits (see new **Supplementary Figure 2A**). Importantly, we show that aptamers KU7 and KU9 bind with low nM affinity to histones, H1, H3, and H4 (major histones implicated in MODS). In contrast, while only aptamer KU7 binds to histone H2B, neither aptamer binds to histone H2A. These data suggest that aptamer KU7 may be indicated for the treatment of sepsis-induced MODS.

- 2) It would be important to determine if the aptamers neutralize both "natural" histones and histones derived from neutrophils induced in vitro to form Nets in the presence or absence of PAD4, which converts arginine to citrulline, which occurs naturally during sepsis. The prediction would be that aptamers would work in both situations, but there are no available data. There are several PAD4 inhibitors that are commercially available and could be used for such studies. These data would be important for an understanding of what the aptamers are doing in vitro and in vivo.*

We agree with the reviewer that it is important to test whether our aptamers bind to histones derived from neutrophils (e.g. neutrophil extracellular traps - NETs). We have added new data to directly answer this question. In new Figure 6, human neutrophils were induced with PMA to induce NETs (new **Figure 6A**). Using confocal microscopy, we demonstrate binding of aptamer

KU7 to human neutrophil-derived NETs . Additionally, we added new data demonstrating that aptamer KU7 inhibits human neutrophil extracellular traps (NETS)-induced cytotoxicity (new **Figure 6B**).

Reviewer #2 (remarks to the author)

The authors describe efforts towards the development of novel aptamers that have high affinity for extracellular histones, which is implicated in the organ dysfunction syndrome, MODS. The first few pages of the manuscript are devoted to the pathology of MODS and limitations of current therapies. The authors make many points in this study. For example, they use 2'-fluoro modified RNA that effectively prevented RNA degradation induced by nuclease in human serum. Also, during the course of the RNA selection, the authors perform both positive and negative screenings in an attempt to impart specificity to binding to histone H3 versus H4 and no other proteins. The RNA aptamers inhibit histone H4-induced platelet aggregation in in vitro experiment, whereas they inhibit IL-6 production and increase of calcium concentration in vivo. The authors test intravenous injection of histone with/without the aptamer into mice where the aptamer increases mice survival, prevents vascular congestion, and improves survival of histone-injected mice.

My major concern is that there is no evidence for specificity of the aptamers. Round 0 and later-round aptamers work almost equally in binding histones, bind the non-cognate histone tightly, and promote platelet aggregation nearly equally. Moreover, binding is not at equilibrium, which may or may not explain the effects. Also, the primer binding sites are involved in aptamer formation. While some physiological effects might be interesting, it seems the same results could be obtained just by using 2'-fluorinated random RNA.

With regards to equilibrium binding, we have now performed binding studies at 5 min, 50 min and 100 mins to allow the reaction to come to equilibrium. We see no statistically relevant differences in binding affinity constants for the aptamers at any of the time points tested, suggesting that the 5 min timepoint is close to equilibrium (new **Supplementary Figure 2B**).

With regards to specificity, binding specificity was engineered by selecting for aptamers that bind to extracellular histones **BUT DO NOT BIND** to other proteins present in human serum. While it is true that most 2'-F-modified RNA aptamers bind to the histones, we see specificity with respect to some of the aptamers binding to different histone isoforms (see new **Supplementary Figure 2A**). For example, aptamers KU7 and KU9 bind with low nM affinity to histones, H1, H3, and H4 (major histones implicated in MODS). In contrast, while only aptamer KU7 binds to histone H2B, neither aptamer binds to histone H2A.

Major Comments

- 1. The authors describe "a leftward shift in the binding curve of the R8 RNA pools compared to the R0 RNA pools for both selections is indicative of enrichment of high-affinity binding RNA sequences" at the first paragraph on Result section. The random RNA pools at the round 0, however, seem like they already have strong affinity for both histone H3 and H4. Is this because histone proteins have non-specific affinity for nucleic acids? The authors should plot their Figure 1B data on a logarithmic x-axis to spread out the data near the origin to help the reader judge any differences. Moreover, logarithmic concentrations are linear in free energy and allow the reader to look for sigmoidal behavior and judge the Hill coefficient. In such a plot, the zero-protein concentration is omitted and can be added with an axis break if desired.*

As suggested by the reviewer, we have now plotted the data in **Figure 1B** on a logarithmic scale (included below). As pointed out by the reviewer, the difference is minimal and suggests that

many aptamers in the library bind to the histone proteins. This is not surprising given the nature of the target protein. The reviewers point is well taken. A key feature of our selection strategy was to identify unique aptamer sequences that bound to histones with high affinity while at the same time did not bind to serum proteins, which is critical for their application in a clinical

Figure 1B

setting.

2. *The above concern is reinforced the aggregometer readings in Supp Fig 3 where R0 aptamers have nearly identical efficacy in preventing aggregation with time as R3/R5 aptamers.*
See comment 1 above.

3. *p7. Specificity of the aptamers is further called into question also by the fact that KU7 and KU9 bind to H3 AND H4 even though they were selected for binding to H3 only.*

The reviewer's point is well taken. The aptamers have some overlapping specificity for histones due to charge interactions. This is advantageous given that a variety of histones are release in the context of MODs. The ideal therapeutic agent will need to be able to address multiple histones species. We would suggest that the combination of high affinity histone binding and lack of binding to serum proteins is the key feature of these aptamers and is critical to their utility in treating MODS.

4. *On a related note, the size of the library, which should be mentioned in the main text, is only 20 nt.*

The aptamer library is composed of a 20 nt variable region flanked by two constant regions (15 and 16 nt in length). The total length of the RNA aptamers derived from this library is 51 nt. We have now indicated this in the text of the manuscript.

5. *p7. This reader does not understand the sentence, "The top three RNA sequences from each selection (H3 selection: KU4 – KU6; H4 selection: KU7 – KU9) were selected based on sequence enrichment between rounds 3 to 5 (positive selection) and rounds 6 to 8 (negative selection)." Why is rounds "3 to 5" followed by "(positive selection)" and "rounds 6 to 8" followed by "(negative selection)". Also, is the first part of this sentence meant for H3 selection and the second part for H4 selection? Also, why the abbreviation "KU"?*

This statement was revised in the text. It now states: “The top three RNA sequences from each selection (H3 selection: KU4 – KU6; H4 selection: KU7 – KU9) were selected based on the following criteria: sequences that increase in abundance during the positive selection rounds (against target histones H3 or H4) but do not decrease in abundance during the negative selection rounds (against BSA and human IgG) (see **Supplementary Table 1** for details on selection conditions).”

The KU abbreviation for aptamer identification was based on the initials of the individual in the laboratory who performed the selection. This is a standard procedure for classifying aptamers in our laboratory.

- The time for equilibration of protein and aptamer was only 5 minutes. At low concentrations of aptamer this is not enough time for binding. e.g. Aptamer was reported in the SPR in Fig 2 to have an on-rate constant of $1.6e5 \text{ M}^{-1}\text{s}^{-1}$. At 1 nM of aptamer, which is in Figure 1D and would be needed for K_d of 1 nM as in Supp Table 2, would need on the order of 6000 sec $[=1/(1.6e5\text{M}^{-1}\text{s}^{-1} * 1e-9\text{M})]$ or 100 min (~2h) to bind to completion. Since this length of time wasn't tested, there is no way to know if the system is at equilibrium. The authors must assure equilibrium by showing they get the same answer with longer times of equilibration. Indeed, this may be the root cause of why binding curves H3 R0 and H3 R8 are essentially superimposable in Fig 1B. As stated above, we have now performed binding studies at 5 min, 50 min and 100 mins to allow the reaction to come to equilibrium. We see no statistically relevant differences in binding affinity constants for the aptamers at any of the time points tested, suggesting that the 5 min timepoint is close to equilibrium (new **Supplementary Figure 2B**).*

Along these lines, In Fig. 1B, Fig. 2 and Supplementary Fig. 2A, the authors should consider appropriate range of higher histone-concentration (i.e. concentrations chosen are too high). In Fig. 1B and Supplementary Fig. 2A, the fraction bound of aptamers (in particular, H3R8, H4R0, KU7 for H3.2 and H4, and KU9 for H4) reaches at plateau at second lowest concentration of histones. In Fig. 2, estimated K_D values are out of the range that they tested, which means the test condition is less quantitative. Appropriate range is in a range of lower histone concentration. We have revised the data accordingly (Supplementary Fig 2A). Concentration of histone for binding experiments include both low and high concentrations to accurately determine K_d , while the concentration of histones for the in vitro functional experiments use concentrations of histones reported in the literature for MODS.

- p7. The authors state that they reasoned that two of the aptamers would be stable but don't explain their reasoning. We have now clarified this statement to indicate that aptamers with greater complementary regions (stems) vs. unpaired regions (loops) are likely to be more structured and stable (see new **Supplementary Figure 1**, theoretical secondary structures for top six histone aptamers and **Supplementary Figure 3**). Based on the predicted secondary structure analysis (new **Supplementary Figure 1**), it is not surprising that aptamers KU7 and KU9 (which have longer stem regions) are more stable than aptamer KU5 (which has a larger unstructured loop region) (**Supplementary Figure 3**).*
- Supp Fig 1. What is the meaning of the different colors? Why are there two structures provided for each sequence? Why are the flanking regions involved in the aptamer? Does the fact that*

they are interacting with the random region and perhaps more importantly with themselves at the base of every structure mean that the random region wasn't long enough?

We have now modified **Supplementary Figure 1** to show a theoretical secondary RNA structure for each of the top 6 histone aptamers derived from the selections. As for other previously reported aptamers, it is not surprising that the constant regions also contribute to the overall folding of the RNA aptamer. From previous experience comparing aptamers libraries with random regions of different lengths, we still see a contribution of the constant regions to overall aptamer binding and function (our unpublished data).

9. *Despite the authors evaluate RNA stability in human serum in Fig. 2B, they don't show the corresponding gel image in KU5 in Supplementary Fig. 2B. Also, what is the reader looking at in Supp Fig 2B? Is this the top of a gel? If so, the degradation down on the gel should be shown. Or is this a slot blot? If so, the retention of various RNA lengths should be provided because there could be complete degradation and still lots of signal.*

We agree with the reviewer and now provide the full gel images (new **Supplementary Figure 3**).

10. *p14. The argumentation for 2'F does not make good chemical sense. It makes sense that 2'F increases duplex stability, by locking in the proper sugar pucker of 2'-exo/3'-endo for an A-form duplex, as the authors state. But the aptamers in the present study bind proteins not other RNAs, and there is no reason a priori to think 2'-exo sugar puckers are needed for this. More likely is that resistance to nuclease-mediated and chemical degradation, which rely on the 2'OH, is offered by the 2'F.*

We agree with the reviewer that nuclease stability is a major outcome of the 2'F modification and have acknowledged this point (see revised Discussion section; page 14, first paragraph). Indeed, there is also substantial evidence in the literature to suggest that the 2'F modification, in addition to its important function in sculpting RNA conformation, plays an underappreciated role in modulating Watson-Crick base pairing strength and potentially π - π stacking interactions (Patra et al., *Angew. Chem. Int. Ed. Engl.* 2012; Pallan et al., *NAR* 2010). In addition, post-selection modifications of the 2'-position of aptamers has been documented to increase binding to the target protein (Mittleberg et al., *RNA Biol.* 2015; Da Rocha Gomes et al, *Bioconjug. Chem.* 2012; Meyer et al., *RNA Biol.* 2014).

11. *p4. It is not wholly accurate to say that aptamers do not aggregate. Aggregation of RNA by base pairing or quadruplex formation is well known.*

We agree with the reviewer and have acknowledged this point.

12. *p5. High affinity (nM Kd) does not always lead to "an excellent reagent". Specificity plays an equally important role as affinity in the process.*

We agree with the reviewer and have acknowledged this point.

13. *p11. The text mentions that mice were treated with KU9 at 30 min and references Fig 5C. But Fig 5C shows KU9 administered at 180 and 260 min only.*

We have now added an arrow to indicate the time of aptamer administration (30 min).

14. *Fig 5A suggests that Vehicle blocks cell survival which doesn't make sense. Are the symbols on Vehicle and Heparin switched?*

The data in Figure 5A represent cell survival in the presence of histones. The data show that Heparin, and aptamers KU7 and KU9 are capable of inhibiting histone-mediated cell death whereas, as expected, vehicle (PBS) has no effect.

15. *Page 10. "Death occurred between one to three hours of aptamer (Fig. 4A and 5B)". This should be Fig. 4A and 5C. Following sentence "Injection of histones by the aptamer treatment (Fig. 4B and 5C)". This should be Fig. 4B and 5D.*

Corrected

16. *p12. Remove comma between "which" and "would".*

Done

17. *In Fig. 3C, CTH concentrations should be represented by molar concentration. it is helpful to figure out molecular ratios between aptamer and CTH.*

Because the CTH histone prep is composed of multiple types of histones it is not possible to determine an actual molar ratio.

18. *The authors should provide unit in Supplementary Table 2.*

Done

19. *Page 10. "Death occurred between one to three hours of aptamer (Fig. 4A and 5B)". This should be Fig. 4A and 5C. Following sentence "Injection of histones by the aptamer treatment (Fig. 4B and 5C)". This should be Fig. 4B and 5D.*

Corrected

20. *p12. Remove comma between "which" and "would".*

Done

Reviewers' Comments:

Reviewer #1:

None

Reviewer #2:

Remarks to the Author:

The authors did a thorough job of addressing my concerns. The two points that remain are as follows.

1.) There just isn't much specificity here WITH RESPECT TO THE RNA. The new plots in Fig 1b provided in response to my point 1 clearly show this, where "many aptamers in the (starting) library bind to the histone proteins (equally well)". The authors do however support the claim that there is specificity WITH RESPECT TO PROTEIN—the aptamers bind histones and not other proteins. I think this point is perhaps not made clearly enough in the paper. Lack of specificity wrt RNA need not be seen as a weakness but even a possible strength. Perhaps a LIBRARY of 2'-fluorinated RNA will be better therapeutics than a single RNA alone, as a pool of different RNAs could evade resistance and provide more ways to provide platelet aggregation. Even non-selected 2'-F RNA could work. I'd like to encourage the authors to address this point, and even emphasize this possible application, should they agree.

2.) The kinetics are still a bit odd. SPR says the on-rate constant is only $10^5 \text{ M}^{-1}\text{s}^{-1}$. Binding should NOT be complete in 5 min for reasons provided in my back-of-the-envelope calculation in the original review. I don't doubt the new results that binding is complete in 5 min, but I now question that the on-rate constant is this slow in serum. The authors should re-evaluate their SPR data and see if k_{on} is calculated correctly (and evaluate error in the value), and if there are no mistakes or slop in the number, they should offer a reason for this discrepancy.

Rebuttal

Once again, we thank the reviewers for their detailed and helpful comments on our manuscript. We have now addressed reviewer 2 remaining concerns and revised the manuscript accordingly. Changes in the manuscript text are highlighted using track changes. Excerpts from the reviewers' critiques are indicated below in *italics*.

Reviewer #2 (remarks to the author)

1. *There just isn't much specificity here WITH RESPECT TO THE RNA. The new plots in Fig 1b provided in response to my point 1 clearly show this, where "many aptamers in the (starting) library bind to the histone proteins (equally well)". The authors do however support the claim that there is specificity WITH RESPECT TO PROTEIN—the aptamers bind histones and not other proteins. I think this point is perhaps not made clearly enough in the paper. Lack of specificity wrt RNA need not be seen as a weakness but even a possible strength. Perhaps a LIBRARY of 2'-fluorinated RNA will be better therapeutics than a single RNA alone, as a pool of different RNAs could evade resistance and provide more ways to provide platelet aggregation. Even non-selected 2'-F RNA could work. I'd like to encourage the authors to address this point, and even emphasize this possible application, should they agree.*

We have now addressed the reviewer's point regarding specificity with respect to protein. See Discussion section, paragraph two. The revised paragraph now states: "In addition, binding of the histone aptamers to multiple histone subtypes could also have a therapeutic advantage by enabling the inactivation of several histone proteins implicated in MODS with one aptamer drug. Similarly, a pool of (unselected) 2' F-modified RNA aptamers, with low affinity for serum proteins, could be used to neutralize all cationic nuclear proteins (including all histones subtypes) release in circulation following extensive cell injury, thereby maximizing therapeutic efficacy.

2. *The kinetics are still a bit odd. SPR says the on-rate constant is only $10^5 M^{-1}s^{-1}$. Binding should NOT be complete in 5 min for reasons provided in my back-of-the-envelope calculation in the original review. I don't doubt the new results that binding is complete in 5 min, but I now question that the on-rate constant is this slow in serum. The authors should re-evaluate their SPR data and see if k_{on} is calculated correctly (and evaluate error in the value), and if there are no mistakes or slop in the number, they should offer a reason for this discrepancy.*

We thank the reviewer for bringing up this point. Below, we have outlined several potential reasons for the discrepancy between the binding constants for the aptamer histones complexes generated by double-filter binding assay or SPR. In addition, we revised the manuscript text (see revised results section) to include these points.

1. One possibility for the discrepancy observed in the binding constants generated using double-filter binding assay and SPR is due to the inherent differences with the two assays used for measuring binding. For example, the double-filter binding assay is performed in solution (e.g. histone and aptamers are free to interact in solution). In contrast, binding with SPR is assessed by immobilizing the ligand (e.g. histones) on a solid surface. Heterogeneity in the orientation and density of the histone molecules displayed on a solid surface is likely to

impact binding measurements obtained with SPR vs. solution-based assays. For the double-filter binding assays, the RNA aptamers and histones were incubated in solution and allowed to reach equilibrium before capturing on a nitrocellulose membrane. K_{on} in solution is diffusion limited. In contrast, K_{on} for SPR (steady state equilibrium assay) is dependent on the density of the histone molecules immobilized on the solid surface and the mass of the RNA aptamer. As observed for our studies, in conditions of high density of the ligand (histone) when the rate of analyte (RNA) binding the ligand (histones) exceeds the rate at which the analyte is delivered to the surface, apparent association rate constants will be lower than the actual K_{on} . Our calculations for association rates are shown in the table below. Theoretical K_{on} values obtained with the double-filter binding assay are approximately 10 fold higher than those obtained using SPR.

	M-1s-1	s-1		s	Mins	Mins	
Kd	Kon	Koff		Half life t1/2	Half Life t1/2	Time for equilibrium	
5.60E-09	1.60E+05	8.96E-04		7.73E+02	1.29E+01	64.5 mins	
5.60E-09	1.60E+06	8.96E-03		7.73E+01	1.29E+00	6.4mins	
5.60E-09	1.60E+07	8.96E-02		7.73E+00	1.29E-01	0.6mins	
Time for equilibrium = 5X t1/2							

- Another difference between the two methods used to assess binding affinity of the RNA aptamers for the histones was the nature of the binding buffers used. For example, the double-filter binding assay was performed in Binding Buffer (20 mM HEPES, 0.15 M NaCl, 2 mM CaCl₂, while the SPR assay was carried out in Binding Buffer + 0.01% Tween-20. The critical micelle concentration (CMC) of Tween-20 is 0.007% at room temperature. Therefore, it is conceivable that the presence of micelles could interfere with binding kinetics.

References

Van Der Merwe, P. Anton. "Surface plasmon resonance." *Protein-ligand interactions: hydrodynamics and calorimetry* 1 (2001): 137-170.

Reviewers' Comments:

Reviewer #2:

Remarks to the Author:

The authors have addressed my remaining concerns. I now find the article acceptable for publication.